# Conditional Learned Reconstruction for Medical Imaging

**Nikita Moriakov**[1,2,*] **George Yiasemis**[1,2,*] **Jan-Jakob Sonke**[1,2] **Jonas Teuwen**[1,2,3]

N.MORIAKOV, G.YIASEMIS, J.SONKE, J.TEUWEN@NKI.NL

[1] *Netherlands Cancer Institute* [2] *University of Amsterdam* [3] *Radboud University Medical Center*

**Editors:** Accepted for publication at MIDL 2026

## Abstract

Medical imaging utilizes a handful of different imaging modalities such as tomography and magnetic resonance (MRI) imaging that require solving an inverse problem to reconstruct an image from the acquired measurements. Reconstruction methods based on learned iterative schemes have been widely explored recently, however, these modalities involve variability in hardware- and protocol-dependent acquisition parameters such as tube current and projection count in case of tomography and acceleration factor or field strength in case of MRI, which are typically not accounted for in the architecture. In this work we propose the framework of conditional learned iterative schemes, where the network weights are explicitly adapted as learned functions of the acquisition parameters. We compare conditional learned iterative schemes to their counterparts without conditioning for both tomography and MRI and demonstrate their effectiveness.

**Keywords:** Inverse Problems, Medical Imaging, Conditional Learned Reconstruction.

## 1. Introduction

Modern medical imaging relies on multiple modalities such as Magnetic Resonance Imaging (MRI) and Computed Tomography (CT). Unlike chest X-ray or digital mammography, MRI and CT acquire measurements that must be transformed into anatomical images through a reconstruction procedure that explicitly incorporates the physics of the acquisition process.

In accelerated MRI, the scanner collects an undersampled version of the signal's Fourier transform, providing only a masked subset of $k$-space. Reconstruction therefore requires estimating the missing data using prior knowledge about the underlying image, as in compressed sensing approaches (Lustig et al., 2007). Acquisition parameters such as acceleration factor, low-frequency sampling density, or field strength vary across protocols and hardware, and these variations directly affect image quality and noise characteristics.

In tomography, the measurements constitute a set of one- or two-dimensional projection images recorded by the X-ray detector as it rotates around the patient, which is the source of photons, positioned on the opposite side. Fan-beam CT refers to the two-dimensional variant where X-rays diverge from the source in a fan shape forming a one-dimensional projection image, while Cone-beam CT (CBCT) describes the three-dimensional variant with X-rays diverging in a broad cone, forming a two-dimensional projection image. The recorded intensities reflect the attenuation coefficients along each ray, and reconstruction seeks to recover these coefficients using classical reconstruction methods such as filtered back-projection (FBP) algorithm (Radon, 1986; Feldkamp et al., 1984; Markoe, 2006) and iterative reconstruction with some form of regularization (Kaipio and Somersalo, 2005). As

---

\* Equal contribution

in MRI, acquisition settings such as tube current and projection count differ across scanners and protocols; e.g., lower tube current reduces photon count and the associated radiation exposure but increases the noise, which in turn affects the ideal regularization strength.

Although classical reconstruction methods remain routinely used, deep learning-based reconstruction approaches have gained traction surpassing classical methods in recent reconstruction challenges (Beauferris et al., 2022; Muckley et al., 2021). A broad spectrum of these approaches adopt learned iterative schemes, inspired by classical iterative methods, while utilizing the measurement operator and/or its adjoint into the architecture. Examples include Learned Primal-Dual (Adler and Öktem, 2018), $\partial$U-net (Hauptmann et al., 2020a), Recurrent Inference Machines (Lønning et al., 2019) and Variational Networks (Hammernik et al., 2017; Yiasemis et al., 2022).

However, the challenge remains that the variability in hardware and acquisition protocols often isn't directly accounted for within the reconstruction network architecture. This omission forces the network to infer acquisition parameters implicitly, leading to potential inaccuracies. Also, training separate models for all possible settings is generally not feasible. This work aims to address these issues by introducing a framework for *conditional learned iterative schemes*, where the model parameters are adapted in a *learned* way to the physical acquisition settings of each individual sample.

Several prior works have explored conditioning neural networks on auxiliary information in medical imaging. A common strategy is feature-wise modulation of activations, where learned affine transformations condition intermediate feature maps on side information, as in FiLM (Perez et al., 2018) and Adaptive Instance Normalization (Huang and Belongie, 2017), with applications to medical image segmentation and representation learning (Lemay et al., 2021; Liu et al., 2022). Another line of work relies on hypernetwork-based conditioning, where a separate network generates the weights of the reconstruction model as a function of acquisition parameters, yielding context-specific models (Ramanarayanan et al., 2023b). More recently, adaptive convolution has been proposed for QSM dipole inversion, where convolution kernels are generated from acquisition geometry parameters within a feed-forward U-Net architecture (Graf et al., 2024). In contrast, we propose a lightweight conditioning of the reconstruction operator via modulated convolutions, where learnable weights are partially modulated given the acquisition parameters. This design is architecture-agnostic and naturally extends beyond reconstruction, enabling principled use of auxiliary acquisition information across a broad range of medical imaging tasks where such metadata is available. Our contributions can be summarized as follows:

- We introduce the concept of modulated convolutions within iterative image reconstruction schemes to facilitate conditional learning.
- We evaluate conditional versus non-conditional learned iterative schemes for accelerated MRI and Cone-beam/Fan-beam CT reconstruction, displaying consistent improvements.

## 2. Background

### 2.1. Inverse problems

Mathematically, addressing an abstract inverse problem typically involves solving an equation expressed as

$$y = \mathcal{A}(x_{\text{true}}) + \varepsilon, \tag{1}$$

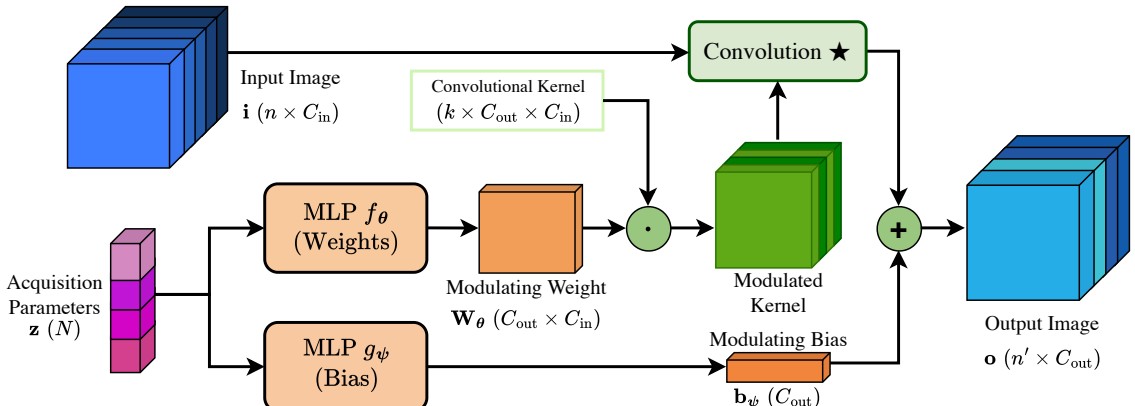

Figure 1: Schematic of the Modulated Convolutional layer. The acquisition parameters $\mathbf{z}$ feed into two separate MLPs, $f_\theta$ and $g_\psi$, to generate a modulating weight $\mathbf{W}_\theta$ and a bias $\mathbf{b}_\psi$. These terms modulate the convolutional kernel $\mathbf{k}$ and the convolution output, respectively, conditioning the process on the physical acquisition parameters.

where $x_{\text{true}} \in X$ represents the true model parameters that we aim to estimate, $y \in Y$ is the observed data, $\mathcal{A} : X \to Y$ denotes the observation operator, and $\varepsilon$ signifies the observation noise. This equation, also referred to as the forward model, establishes the relationship between $x_{\text{true}}$ and $y$. A solution to (1) can be approached by explicitly minimizing the negative data log-likelihood $\mathcal{L}$ to identify the maximum likelihood estimator. This entails estimating $x_{\text{true}}$ by

$$\hat{x} := \operatorname*{argmin}_{x \in X} \mathcal{L}(\mathcal{A}(x), y). \tag{2}$$

However, in the case of ill-posed inverse problems or when noise is present, this can lead to noise overfitting. To prevent this, variational regularization can be applied estimating $x_{\text{true}}$

$$\text{as} \quad \hat{x}_\lambda := \operatorname*{argmin}_{x \in X} \left( \mathcal{L}(\mathcal{A}(x), y) + \lambda \mathcal{G}(x) \right), \tag{3}$$

where $\mathcal{G} : X \to \mathbb{R}$ denotes a regularization functional describing prior knowledge about $x_{\text{true}}$ such as smoothness or sparsity, and $\lambda > 0$ is the regularization parameter. For many classical inverse problems arising in image reconstruction, iterative methods exist that allow to approximate (3) numerically, while different strategies have been proposed for picking the optimal value of $\lambda$. For instance, the Morozov discrepancy principle (Morozov, 1966; Kaipio and Somersalo, 2005) tightly bounds the noise at the true solution by $\mathcal{L}(\mathcal{A}(x_{\text{true}}), y) \leq \epsilon$ and then selects $\lambda$ such that $\mathcal{L}(\mathcal{A}(\hat{x}_\lambda), y) \approx \epsilon$. It's important to note that the choice of $\lambda$ and the regularization functional $\mathcal{G}$ can influence the parameters of the iterative scheme used to solve (3), such as the step size or total iteration count.

## 2.2. Accelerated MRI Reconstruction

In Accelerated MRI Reconstruction, the goal is to reconstruct an image $x \in \mathbb{C}^n$ from sparsely sampled multi-coil ($n_c > 1$) $k$-space data $\tilde{y} \in \mathbb{C}^{n \times n_c}$. This process relies on knowledge of coil sensitivities $\mathbf{S} = (\mathbf{S}_1, \ldots, \mathbf{S}_{n_c}) \in \mathbb{C}^{n \times nc}$, which reflect each coil's spatial sensitivity. The forward model is described by a linear operator $\mathcal{A}_{\Theta,\mathbf{S}} : \mathbb{C}^n \to \mathbb{C}^{n \times n_c}$, defined as:

$$\tilde{y} = \mathcal{A}_{\Theta,\mathbf{S}}(x) := \mathbf{U}_\Theta \circ \mathcal{F} \circ \mathcal{E}_{\mathbf{S}}(x), \tag{4}$$

combining undersampling via an operator $\mathbf{U}_\Theta$, the two-dimensional Fast Fourier transform (FFT) $\mathcal{F}$ and the coil-encoding operator $\mathcal{E}_{\mathbf{S}} : \mathbb{C}^n \to \mathbb{C}^{n \times n_c}$, which maps an image to individual coil images using $\mathbf{S}$. Details on $\mathbf{U}_\Theta$ are provided in Appendix A.1.

### 2.3. Cone-beam CT

Cone-beam CT reconstruction seeks to recover the spatially varying X-ray attenuation coefficients $x \in X \subset \mathbb{R}^3$ from noisy projection measurements $y$. For a monochromatic X-ray source, the forward model is defined by the projection operator $\mathcal{P}$ (defined in Appendix A.2), which integrates $x$ along rays from the source to detector elements. Assuming a Poisson noise model following the Beer–Lambert law, the data acquisition is described by

$$y = \texttt{Poisson}(I_0 \cdot e^{-\mathcal{P}x}), \tag{5}$$

where $I_0$ denotes the unattenuated photon count. Higher photon and projection counts reduce image noise but increase radiation dose. The reconstruction problem is to recover $x$ given $y$.

### 2.4. Fan-beam CT

Fan-beam CT is the two-dimensional analogue of CBCT, where the domain is $X \subset \mathbb{R}^2$ and the projection operator $\mathcal{P}$ is defined by line integrals along rays in the imaging plane. The full geometric description of the source trajectory, detector parametrization, and projection mapping is provided in Appendix A.3. The noisy acquisition model follows (5).

## 3. Methods: Conditional learned iterative schemes

In order to efficiently adapt an arbitrary learned iterative reconstruction scheme to the current physical acquisition parameters, we propose to *modulate* the convolutional filter intensities by a learned function of the acquisition parameters. To that end, we introduce the *Modulated Convolutional* layer in Section 3.1 which is used in place of the traditional convolutional layer employed in iterative reconstruction schemes.

Our motivation for modulation is based on the discussion about the optimal choice of regularization parameters in Section 2.1, and in particular the Morozov discrepancy principle. We note that when working with a heterogeneous dataset with varying noise characteristics, such as tube current in tomography or field strength and acceleration parameter in MRI, the optimal amount of regularization would likely vary as well. However, if the learned iterative scheme is not informed about such variation, it would be forced to try to estimate noise characteristics from the image data directly, which can complicate the learning process leading to sub-optimal results and poor generalization to other acquisition schemes.

### 3.1. Modulated Convolution

- Let $\mathbf{i} \in \mathbb{R}^{n \times C_{\text{in}}}$ denote an input image with spatial support $n = n_1 \times n_2$ in 2D or $n = n_1 \times n_2 \times n_3$ in 3D, and $C_{\text{in}}$ the number of input channels.

- Let $\mathbf{k} \in \mathbb{R}^{k \times C_{\text{out}} \times C_{\text{in}}}$ represent a convolutional kernel, where $k = k_1 \times k_2$ in 2D or $k = k_1 \times k_2 \times k_3$ in 3D, and $C_{\text{out}}$ the number of output channels.

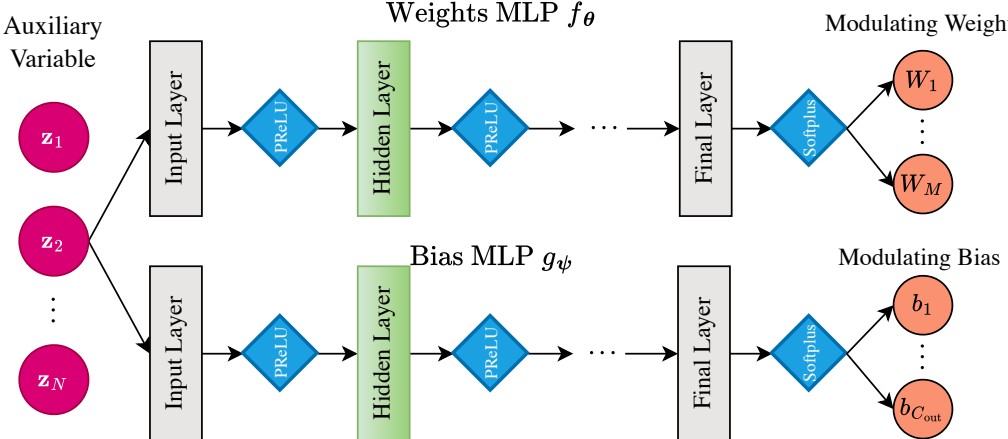

Figure 2: Modulator architecture: The auxiliary variable $\mathbf{z} \in \mathbb{R}^N$, encoding acquisition characteristics, is passed through two distinct sequences of linear layers and activations to produce the modulation weight $\mathbf{W} \in \mathbb{R}^M$ with $M = C_{\text{out}} \times C_{\text{in}}$ and bias $\mathbf{b} \in \mathbb{R}^{C_{\text{out}}}$.

• Let $\mathbf{o} \in \mathbb{R}^{n' \times C_{\text{out}}}$ be the resulting output image, with $n'$ defined analogously to $n$.

Additionally, let $\mathbf{z} \in \mathbb{R}^N$ denote an auxiliary vector containing the acquisition parameters. Utilizing the aforementioned notations, we define the modulated convolution as:

$$\mathbf{o}_m = \sum_{k=0}^{C_{\text{in}}-1} ((\mathbf{W}_{\boldsymbol{\theta}})_{m,k} \cdot \mathbf{k}_{m,k}) \star \mathbf{i}_k + (\mathbf{b}_{\boldsymbol{\psi}})_m, \quad \mathbf{W}_{\boldsymbol{\theta}} = f_{\boldsymbol{\theta}}(\mathbf{z}) \in \mathbb{R}^{C_{\text{out}} \times C_{\text{in}}}, \quad \mathbf{b}_{\boldsymbol{\psi}} = g_{\boldsymbol{\psi}}(\mathbf{z}) \in \mathbb{R}^{C_{\text{out}}}. \quad (6)$$

where $\cdot$ denotes scalar multiplication and $\star$ represents the cross-correlation operation, $m = 1, \cdots, C_{\text{out}}$, and $f_{\boldsymbol{\theta}}$ and $g_{\boldsymbol{\psi}}$ refer to the components of the modulation model, implemented as (trainable) multi-layer perceptrons (MLPs). These MLPs take the auxiliary vector $\mathbf{z}$ as input and produce a modulating weight that adjusts the convolutional kernel and a bias tailored to the acquisition parameters. Consequently, the convolutional process becomes conditioned on the auxiliary variable, enhancing its adaptability. Note that the conventional (*unmodulated*) convolution can be obtained by setting $f_{\boldsymbol{\theta}} = \mathbf{1} \in \mathbb{R}^{C_{\text{out}} \times C_{\text{in}}}$ and $g_{\boldsymbol{\psi}} = \boldsymbol{\psi} \in \mathbb{R}^{C_{\text{out}}}$. Each MLP is structured with linear layers followed by parametric ReLU (PReLU) activation functions, and a Softplus activation is applied after the final layer.

In Fig. 1 we provide a depiction of a modulated convolution while the architecture of the modulation model is illustrated in Fig. 2. A more generalized form of our proposed conditioning method, for different types of modulation, is detailed in Appendix B.1.

## 3.2. Modulated Transposed Convolution

Many convolutional-based models, particularly those with an encoder-decoder framework like U-Net (Ronneberger et al., 2015), utilize a combination of convolutions within the encoder and transposed convolutions in the decoder. For example, in our accelerated MRI Reconstruction and Cone-beam CT reconstruction experiments, we employ both vSHARP and $\partial$U-Net, which incorporate transposed convolutions. Building upon our concept of

Modulated Convolution, we extend this approach to introduce *Modulated Transposed Convolutions*. This adaptation involves modulating the transposed convolution kernels and biases through (6), using the auxiliary variable input. This method ensures that both encoding and decoding processes in our models are modulated.

### 3.3. Deep Learning Reconstruction Backbones

Our modulation mechanism is architecture-agnostic and can be incorporated into a wide range of learned iterative schemes. In this work, we evaluate it within three representative backbones for each considered modality:

**Iterative ADMM DL-based Accelerated MRI Reconstruction** In our experiments, we adopt an ADMM-based unrolled reconstruction framework in which each iteration alternates a data-consistency update with a learned denoising block. For static 2D reconstruction experiments, we use a 2D reconstruction model operating on individual slices, while for dynamic 2D reconstruction experiments we employ a 3D (2D+time) model that jointly processes spatial and temporal dimensions. All convolutional and transposed-convolutional layers within the learnable components are replaced by the proposed modulated convolutions, enabling the network to adjust its behaviour according to the acquisition parameters of each sample. The full set of update equations, initialization strategy, sensitivity-map refinement module, and network architectures follow the vSHARP formulation (Yiasemis et al., 2024b). Complete mathematical details and implementation specifics are provided in Appendix B.2.1.

**∂U-net** For CBCT we adopt ∂U-net (Hauptmann et al., 2020b), a multi-scale learned iterative scheme operating purely in the image domain. Modulation is applied to every convolutional block across all resolution levels, allowing the model to track changes in acquisition conditions. A complete description of the hierarchical structure and the initialization procedure appears in Appendix B.2.2.

**Learned Primal-Dual Reconstruction** For 2D CT we use the Learned Primal–Dual (LPD) algorithm (Adler and Öktem, 2018), which alternates image-space (primal) and projection-space (dual) updates connected via differentiable projection and backprojection operators. All convolutional layers within both primal and dual updates are replaced by modulated convolutions. Architectural and algorithmic specifics are provided in Appendix B.2.3.

## 4. Experiments

### 4.1. Comparison and Ablation Studies

To assess our modulation method, we conducted experiments across all considered applications using two setups: **(i)** conventional convolutions (No MOD), and **(ii)** modulated convolutions as in Sec. 3.1, including possible configurations: MOD S, MOD M, and MOD L, corresponding to input/output feature sizes of (32, 8), (32, 16), and (32, 32), respectively.

In addition, for MRI experiments where instance normalization is used within the U-Net convolutions, we evaluate an adaptive instance normalization variant, in which all instance normalization layers are replaced by adaptive instance normalization conditioned

via MLPs with 32 and 16 hidden features (AdaIn M). To assess our method further, we apply our proposed modulation only at the input of the network, by modulating the very first convolutional block in the U-Net encoder, while keeping all subsequent encoder, bottleneck, decoder, and output layers unmodulated, thereby substantially reducing the number of additional parameters introduced by modulation (MOD M - inp-only).

## 4.2. Quantitative Analysis

For our quantitative comparative analysis, we utilized established metrics in image processing to evaluate the performance of our experiments. These metrics include the Structural Similarity Index Measure (SSIM), peak Signal-to-Noise Ratio (pSNR), Normalized Mean Squared Error (NMSE) specifically for Accelerated MRI reconstruction accuracy, and Mean Average Error (MAE/L1) for CT or CBCT reconstruction. The mathematical formulations for these metrics can be found in Appendix C.1.

## 4.3. Accelerated MRI Reconstruction

### 4.3.1. DATASETS

We evaluate our method using two distinct datasets for 2D reconstruction. Specifically, we utilized the prostate (Tibrewala et al., 2023) and knee (Zbontar et al., 2019) fastMRI datasets which comprise raw fully-sampled $k$-space data. The prostate data contain T2-weighted scans with 10–30 coils; the knee data comprise coronal proton-density–weighted scans acquired with 16 coils. For the prostate dataset, we used 218 subjects (6,647 slices) for training, 48 (1,462 slices) for validation, and 46 (1,399 slices) for testing. For the knee dataset, we used 973 volumes (34,742 slices) for training, 99 (3,573 slices) for validation, and 100 (3,562 slices) for testing. Our training involved retrospective undersampling of the data, while utilizing the fully-sampled measurements for loss calculation.

In addition, we used the Cardiac MRI Reconstruction 2025 (Xu, 2025) training dataset for 2D dynamic reconstruction experiments. This dataset contains multi–field strength data acquired at both 1.5T and 3T and includes multiple acquisition sequences, with data acquired using 10 coils. We split the data into 444 4D scans (1,970 2D+time series) for training, 74 scans (331 2D+time series) for validation, and 76 scans (275 2D+time series) for testing, ensuring a balanced distribution of 1.5T and 3T acquisitions across splits.

### 4.3.2. UNDERSAMPLING

To simulate various acceleration factors $(R)$, we applied undersampling to our initially fully-sampled dataset, carefully preserving a specific fraction $(r_{\mathrm{acs}})$ of the data in the autocalibration region for each factor. Importantly, these two parameters $R$ and $r_{\mathrm{acs}}$, served as auxiliary variables for model modulation. For training, we randomly selected $R$ within $[4, 16]$, favoring higher acceleration factors (e.g., factors near 16 were chosen four times as often as those near 4) through a continuous triangular distribution (see Appendix C.2). The $r_{\mathrm{acs}}$ values were randomly picked from a uniform distribution in the range $[0.02, 0.08]$. In the testing phase, the models were evaluated at predefined $R$ values (4, 6, 8, 10, 12, 14, and 16) and their corresponding $r_{\mathrm{acs}}$ values (0.08, 0.06, 0.04, 0.035, 0.03, 0.025, and 0.02,

respectively). We adopted an equispaced undersampling approach, as it aligns with common practices in DL-based MRI reconstruction and offers straightforward implementation for clinical applications.

### 4.3.3. Modulation Auxiliary Variables

To modulate our convolutional networks in the Accelerated MRI Reconstruction setup, we utilize as auxiliary variables the acceleration factor $(R)$ of each sample, as well as the ACS fraction $(r_{acs})$ as defined in Appendix A.1. Where available and varied in the datasets, we include the field strength $(F)$ of the sample as an auxiliary variable. More precisely, we use:

$$\mathbf{z} = \log([R, \, 100 \cdot r_{acs}]) \in \mathbb{R}^2 \quad \text{or} \quad \mathbf{z} = \log([R, \, 100 \cdot r_{acs}, \, F]) \in \mathbb{R}^3. \tag{7}$$

### 4.3.4. Training and Optimization Strategy

Models were implemented in PyTorch (Paszke et al., 2017) and optimized using Adam (Kingma and Ba, 2017) with $(\beta_1, \beta_2) = (0.9, 0.999)$ and $\epsilon = 1e-8$. Training was performed on NVIDIA A100 or H100 GPUs, using batch sizes of 2 or 1 for static and dynamic reconstruction, respectively. Static models were trained for 150,000 iterations, while dynamic models were trained for 80,000 iterations. The learning rate schedule linearly increased from $6.7e-4$ to $2e-3$ over the first 1,000 iterations, followed by a 20% decay every 30,000 iterations. Across all experiments, random data augmentations were applied during training, including cropping, flipping, and rotation, to improve robustness and learning efficacy.

The vSHARP reconstruction models followed the architectural and loss design choices of (Yiasemis et al., 2024b), using multi-scale U-Nets for denoising and sensitivity estimation, with task-specific configurations for static and dynamic reconstruction. A dual-domain loss combining image- and $k$-space-based terms was employed. Full architectural details, augmentation specifications, and hyperparameter choices are provided in Appendix C.3.1.

### 4.3.5. Results

Metrics are calculated between the magnitude of the ground truth image and the magnitude of the predicted image. Note that for both datasets we compute the quantitative results on the central $320 \times 320$ reconstructed image region. The average quantitative SSIM and pSNR results for 2D reconstruction are detailed in Tab. 1 (for $R = 4 - 10$) and Tab. S1 (for $R = 12 - 16$) and NMSE in Tab. S2 (NMSE). The results for the 2D dynamic reconstruction are provided in Tab. 2. In overall, our findings reveal a consistent trend: the models equipped with modulated convolutions consistently outperform their non-modulated counterparts, showcasing superior performance in both prostate and knee dataset reconstructions.

For the knee dataset, MOD M emerges as the top performer on average, though it's noteworthy that all modulation variants (MOD S, M, and L) outperform the baseline models, with only three exceptions observed at specific acceleration factors (R=6 for pSNR and R=8 for SSIM with MOD S, R=4 with MOD M - inp-only). Conversely, in the prostate dataset, the non-modulated models showed slightly better performance than MOD S and MOD M in certain cases ($R$ =8,10,12). However, MOD L consistently surpassed the performance of the non-modulated models.

| | Prostate Dataset | | | | | | | |
|---|---|---|---|---|---|---|---|---|
| | Acceleration Factor ($R$) / ACS fraction ($r_{acs}$) | | | | | | | |
| **Method** | 4 / 0.08 | | 6 /0.06 | | 8 / 0.04 | | 10 / 0.035 | |
| | SSIM | pSNR | SSIM | pSNR | SSIM | pSNR | SSIM | pSNR |
| No MOD | 0.9232 | 36.54 | 0.8855 | 34.61 | 0.8591 | 33.28 | 0.8329 | 32.25 |
| MOD S | 0.9237 | 36.99 | 0.8864 | 34.77 | 0.8577 | 33.19 | 0.8309 | 32.16 |
| MOD M | 0.9244 | 36.92 | **0.8871** | 34.76 | 0.8562 | 33.26 | 0.8316 | 32.25 |
| MOD L | **0.9249** | **37.01** | 0.8867 | **34.77** | **0.8610** | **33.40** | **0.8352** | **32.36** |
| | Knee Dataset | | | | | | | |
| | Acceleration Factor ($R$) / ACS fraction ($r_{acs}$) | | | | | | | |
| **Method** | 4 / 0.08 | | 6 /0.06 | | 8 / 0.04 | | 10 / 0.035 | |
| | SSIM | pSNR | SSIM | pSNR | SSIM | pSNR | SSIM | pSNR |
| No MOD | 0.9061 | 38.34 | 0.8894 | 37.08 | 0.8766 | 35.81 | 0.8634 | 34.67 |
| AdaIn M | 0.9055 | 38.50 | 0.8877 | 36.95 | 0.8736 | 35.66 | 0.8605 | 34.64 |
| MOD S | 0.9072 | 38.40 | 0.8896 | 36.98 | 0.8761 | 35.82 | 0.8635 | 34.82 |
| MOD M (inp-only) | 0.9080 | 38.29 | 0.8906 | 37.10 | 0.8770 | 35.83 | 0.8647 | 34.85 |
| MOD M | **0.9096** | **38.71** | **0.8926** | **37.22** | **0.8802** | **36.06** | **0.8679** | **35.05** |
| MOD L | 0.9083 | 38.60 | 0.8908 | 37.21 | 0.8774 | 36.00 | 0.8643 | 34.93 |

Table 1: Quantitative results (SSIM and pSNR) for accelerated MRI reconstruction.

| | Cardiac Dataset | | | | | | | | | | | |
|---|---|---|---|---|---|---|---|---|---|---|---|---|
| | Acceleration Factor ($R$) / ACS fraction ($r_{acs}$) / Field Strength ($F$) | | | | | | | | | | | |
| **Method** | 4/0.08/1.5T | | 6/0.06/1.5T | | 8/0.04/1.5T | | 4/0.08/3T | | 6/0.06/3T | | 8/0.04/3T | |
| | SSIM | pSNR | SSIM | pSNR | SSIM | pSNR | SSIM | pSNR | SSIM | pSNR | SSIM | pSNR |
| No MOD | 0.8971 | 32.65 | 0.8698 | 30.90 | 0.8359 | 28.93 | 0.9083 | 34.04 | 0.8817 | 32.62 | 0.8514 | 30.87 |
| MOD S | **0.9114** | **33.76** | **0.8791** | **31.55** | **0.8426** | **29.32** | **0.9199** | **35.20** | **0.8869** | **32.91** | **0.8542** | **31.13** |

Table 2: Quantitative results for dynamic accelerated MRI reconstruction.



Figure 3: Example from the knee dataset from accelerated MRI experiments for $R = 6$ and $r_{acs} = 0.06$.

To better contextualize the results, we additionally report the parameter count and inference time of all evaluated models in Tab. S4. This analysis reveals that while larger modulation variants such as MOD M and L substantially increase the number of learnable parameters, their inference times remain comparable to the non-modulated baseline.

A key insight from our comprehensive evaluation in the accelerated MRI reconstruction context is the more pronounced improvement in reconstruction metrics offered by modulated methods over non-modulated ones, especially at higher acceleration factors. Another noteworthy observation is that while MOD M achieves the strongest overall performance, MOD

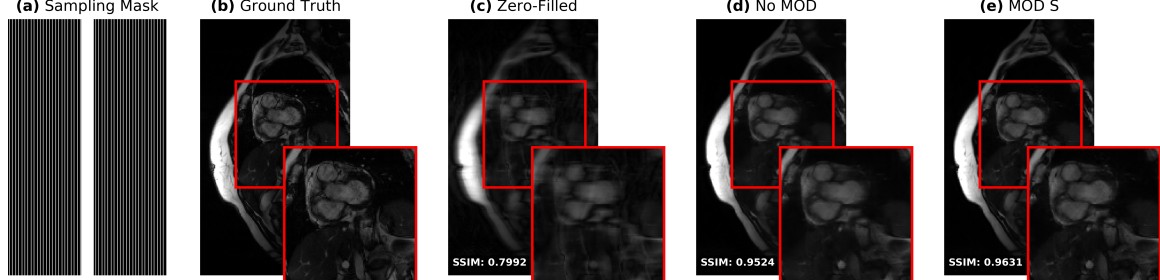

Figure 4: Example from the cardiac dataset from dynamic accelerated MRI experiments for $R = 4$, $r_{\text{acs}} = 0.08$ and $F = 3$T.

M with input modulation only also consistently outperforms the non-modulated baseline at a substantially lower parameter count (see Tab. S4),

For qualitative assessment in Figures 3 and 4 we depict example reconstructions from the knee and cardiac test datasets at $R = 16, r_{\text{acs}} = 0.02$ and $R = 4, r_{\text{acs}} = 0.08, F = 3$, demonstrating the advantage of modulation.

### 4.4. Computed Tomography

#### 4.4.1. DATASETS

For the Cone-beam CT experiments, we used an internal dataset of 424 diagnostic thorax CT scans with isotropic spacing of 1 mm that were dowsampled to 2 mm resolution. The dataset was split into a training set of 260 scans, a validation set of 22 scans and a test set of 142 scans. We simulated a clinical acquisition geometry for a Linac-integrated CBCT scanner from Elekta AB, Stockholm, Sweden(Létourneau et al., 2005) with a medium field-of-view setting, offset detector, a full $2\pi$ scanning trajectory and either (a) 64 projections to simulate low projection count observed in e.g. phase-resolved 4D CBCT reconstruction or (b) a variable projection count $N_{\text{proj}} \in [237, 720]$ to simulate variability observed in e.g. pelvic CBCT acquisitions. The source-isocenter distance was set to 1000 mm and the isocenter-detector plane distance was set to 536 mm. The detector was offset by 115 mm to the side (Sen Sharma et al., 2014) to give an increased Field of View. Square detector panel with a side of 409.6 mm and $256 \times 256$ pixel matrix was used.

For the Fan-beam CT, we used a subset of the Mayo Clinic dataset for the AAPM Low Dose CT Grand Challenge (McCollough, 2016), which was split into training (2961 slices), validation (358 slices) and test (1618 slices) sets. Slices belonging to each subject were assigned to exactly one of the train/validation/test folds. To simulate Fan-beam CT acquisitions, we implemented a fan-beam geometry with source-isocenter and isocenter-detector distances set to 500 mm. Detector size was set to 720 mm with 1000 pixels.

#### 4.4.2. TRAINING AND OPTIMIZATION DETAILS

**Model Optimization**    Models were developed in PyTorch, using Adam with parameters $(\beta_1, \beta_2) = (0.9, 0.999)$ and $\epsilon$=1e-8. Experiments were carried out on NVIDIA Quadro 8000 GPUs, with a batch size of 8 for Cone-beam CT (using gradient accumulation) and batch size of 16 for Fan-beam CT. For CBCT experiments, plateau learning rate scheduler with

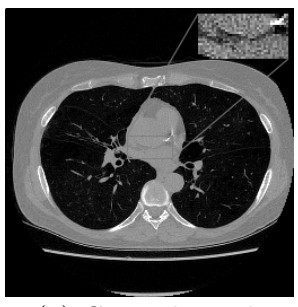
(a) Ground Truth

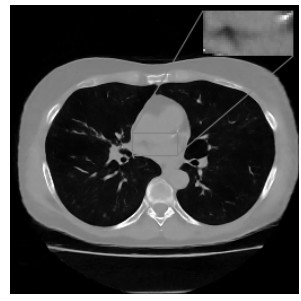
(b) No Modulation

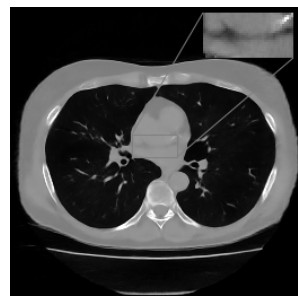
(c) With Modulation

Figure 5: Axial slice of thorax CT, 64 projections, HU range $[-1000, 800]$ and $[-150, 250]$ for the ROI.

linear warm-up during first 130 iterations and evaluation after every 130 iterations was used, learning rate was reduced by a factor of two if no improvements was observed after 5 evaluations. For Fan-beam CT, warm-up period was 1k iterations and evaluation took place every 10k iterations. The training was terminated after learning rate became smaller that $10^{-5}$, which resulted in iteration count between 33k and 34k for CBCT, and 700k and 850k for the Fan-beam CT models. Model with the best MAE evaluation metric was tested.

**Random Augmentations**  For the CBCT experiments, we randomly augmented the volumes by flipping left/right and top/bottom sides of the patient. For the Fan-beam CT experiment, we randomly augmented the slices by flipping the left/right side of the patient.

**Reconstruction Model Hyperparameter and Loss Function Choice**  Our implementation of $\partial$U-Net relies on the open-source implementation (Hauptmann et al., 2020b) from the authors, where the base filter count was increased from 12 to 32 to increase expressive power but fit into the memory budget. We replaced batch normalization layers with instance normalization layers, since batch normalization resulted in unstable convergence. Our implementation of the Learned Primal-Dual method replicates the original implementation and consists of 10 primal and 10 dual cells, each primal/dual cell being a stack of 3 convolutional layers with 32 channels in the first and the second convolutional layer. To train both LPD and $\partial$U-net, we utilised Mean Absolute Error as the loss function.

### 4.4.3. MODULATION AUXILIARY VARIABLES

To modulate our convolutional networks in the Fan-beam CT setup, we utilize photon count $I_0$ as an auxiliary variable of each sample. More precisely, we let

$$\mathbf{z} := \log([I_0]) \in \mathbb{R}, \tag{8}$$

where $I_0$ was sampled from a triangular distribution (see Appendix C.2) supported on the photon count range of $[2.5k, 40k]$ with 4 times higher density at $2.5k$ compared to $40k$. For the Cone-beam CT experiment with variable photon count, we use triangular distribution supported on $[10k, 50k]$ with 4 times higher density at $10k$ compared to $50k$. For the Cone-beam CT experiment with variable projection count, we let

$$\mathbf{z} := \log([\mathrm{N_{proj}}]) \in \mathbb{R}, \tag{9}$$

| | Thorax Dataset | | | | | | | | | |
|---|---|---|---|---|---|---|---|---|---|---|
| **Method** | $I_0 = 10$k | | $I_0 = 20$k | | $I_0 = 30$k | | $I_0 = 40$k | | $I_0 = 50$k | |
| | MAE | pSNR | MAE | pSNR | MAE | pSNR | MAE | pSNR | MAE | pSNR |
| No MOD | 62.36 | 31.02 | 60.66 | 31.36 | 59.90 | 31.53 | 59.43 | 31.63 | 59.11 | 31.71 |
| MOD S | **61.48** | **31.17** | **59.78** | **31.52** | **59.05** | **31.68** | **58.61** | **31.78** | **58.32** | **31.85** |

Table 3: Quantitative results (MAE and pSNR) for variable photon count Cone-beam CT experiments.

| | Thorax Dataset | | | | | |
|---|---|---|---|---|---|---|
| **Method** | $N_{proj} = 237$ | | $N_{proj} = 478$ | | $N_{proj} = 720$ | |
| | MAE | pSNR | MAE | pSNR | MAE | pSNR |
| No MOD | 28.02 | 37.83 | 26.22 | 38.54 | 25.68 | 38.75 |
| MOD S | **26.65** | **38.44** | **24.77** | **39.27** | **24.00** | **39.59** |

Table 4: Quantitative results (MAE and pSNR) for variable projection count Cone-beam CT experiments.

where the projection count $N_{proj}$ is sampled from $[237, 720]$ uniformly at random and the photon count is kept constant at $30k$.

### 4.4.4. RESULTS

MAE is calculated between attenuation arrays converted to HU, while pSNR is computed for attenuation values directly. Results of the Cone-beam CT experiment for variable photon count are provided in Tab. 3 and for variable projection count in Tab. 4. Results of the Fan-beam CT experiments are presented in Tab. S5. Overall, we observe consistent improvement of $\partial$U-net model equipped with modulated convolution over the non-modulated counterpart, even though we are using the most compact version of the modulator in the CBCT experiment. We present example axial slices from the test set with photon count $I_0 = 10k$ and 64 projections in Fig. 5, showing that the modulated network resolves soft tissue details better. In the Fan-beam CT experiment, we observe that the modulated versions of LPD also generally outperform the non-modulated baseline, however, the degree of improvement is small. We conjecture that this can be a consequence of LPD being able to 'learn' the amount of noise from noisy projections, since, unlike $\partial$U-net, dual blocks of LPD have direct access to the projection data.

## 5. Conclusion

In this study, we have introduced a framework for conditional learned iterative schemes, utilizing a proposed convolution modulation strategy to adjust network parameters based on physical acquisition settings. Our comprehensive evaluation in the contexts of accelerated MRI and CT reconstruction tasks demonstrates the potential of conditional learned iterative schemes to enhance reconstruction quality. Nevertheless, the extent of improvements introduced by the proposed conditioning may vary by task and chosen iterative architecture. An extended discussion is provided in Appendix D.

## Acknowledgments

This work was supported by institutional grants of the Dutch Cancer Society and of the Dutch Ministry of Health, Welfare and Sport. The authors acknowledge the Research High Performance Computing (RHPC) facility of the Netherlands Cancer Institute (NKI) for the computational resources.

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

# Conditional Learned Reconstruction for Medical Imaging – Supplementary Material

## Appendix A. Background

### A.1. Accelerated MRI Reconstruction

#### A.1.1. SUBSAMPLING OPERATOR AND SAMPLING PARAMETERS

Given an index set $\Theta \subset \Omega = \{1, \ldots, n\}$, the undersampling operator $\mathbf{U}_\Theta : \mathbb{C}^n \to \mathbb{C}^n$ retains entries indexed by $\Theta$ and sets all others to zero:

$$(\mathbf{U}_\Theta(w))_i = \begin{cases} w_i, & i \in \Theta, \\ 0, & i \notin \Theta, \end{cases} \qquad i = 1, \ldots, n. \tag{10}$$

The acceleration factor is defined as

$$R = \frac{|\Omega|}{|\Theta|} = \frac{n}{|\Theta|}, \tag{11}$$

and is inversely proportional to the number of acquired $k$-space samples. Sensitivity maps are typically estimated from a fully-sampled central region of $k$-space known as the auto-calibration signal (ACS), denoted $\Theta_{\mathrm{acs}} \subset \Theta$, with ACS ratio

$$r_{\mathrm{acs}} = \frac{|\Theta_{\mathrm{acs}}|}{n}. \tag{12}$$

These parameters determine the sampling pattern and partial Fourier coverage used in the forward model.

### A.2. CBCT Acquisition Geometry and Projection Operator

For monochromatic X-ray energy, the attenuation coefficient at spatial location $z \in X \subset \mathbb{R}^3$ is denoted by $x(z) \in \mathbb{R}_{\geq 0}$. The X-ray source follows a circular trajectory parameterized by $\gamma : [0, 1] \to \mathbb{R}^3$, and the detector is described by a family of planes $Y(t)$, each identified with $\mathbb{R}^2$. For detector coordinate $u \in Y(t)$, let $l_{t,u}$ be the line segment from $\gamma(t)$ to $u$. The projection operator is

$$\mathcal{P}(x)(t, u) = \int_{l_{t,u}} x(z) \, dz, \tag{13}$$

mapping functions on $X$ to functions on $[0, 1] \times \mathbb{R}^2$. The adjoint operator $\mathcal{P}^*$ is the back-projection operator.

### A.3. Fan-beam CT Geometry

Fan-beam CT is a two-dimensional analogue of CBCT with domain $X \subset \mathbb{R}^2$. The X-ray source trajectory is $\gamma : [0, 1] \to \mathbb{R}^2$, and the detector is parameterized by lines $Y(t)$. For $u \in Y(t)$, $l_{t,u}$ denotes the line from $\gamma(t)$ to $u$. Using this notation, the projection operator is given by

$$\mathcal{P}(x)(t, u) = \int_{l_{t,u}} x(z) \, dz,$$

mapping $X$ to $[0, 1] \times \mathbb{R}$.

## Appendix B. Methods

### B.1. Generalized Modulated Convolution

This section defines the *Generalized Modulated Convolution*, a broader formulation of the modulated convolution introduced in (6) and discussed in Section 3.1. Using the same notation, the operation is written as

$$\mathbf{o}_m = \sum_{k=0}^{C_{\text{in}}-1} \left( (\mathbf{W}_{\boldsymbol{\theta}})_{m,k} \otimes \mathbf{k}_{m,k} \right) \star \mathbf{i}_k + (\mathbf{b}_{\boldsymbol{\psi}})_m, \quad m = 1, \cdots, C_{\text{out}}, \tag{14}$$

where $\otimes$ denotes the tensor product, and $\mathbf{W}_{\boldsymbol{\theta}}$ is the generalized modulation weights, produced by

$$\mathbf{W}_{\boldsymbol{\theta}} = f_{\boldsymbol{\theta}}(\mathbf{z}) \in \mathbb{R}^M, \tag{15}$$

responsible for fully conditioning the convolution weight on $\mathbf{z}$. Different choices of $M$ allow the method to express a range of modulation strategies:

- $M = k_1 \times k_2$ (2D) or $M = k_1 \times k_2 \times k_3$ (3D) for kernel-only modulation.

- $M = C_{\text{out}} \times C_{\text{in}}$ for feature (channel-wise) modulation, as used in Sec. 3.1.

- $M = (k_1 \times k_2 \times C_{\text{in}})$ or $(k_1 \times k_2 \times C_{\text{out}})$ in 2D, and $M = (k_1 \times k_2 \times k_3 \times C_{\text{in}})$ or $(k_1 \times k_2 \times k_3 \times C_{\text{out}})$ in 3D, for partial feature modulation over input or output channels.

### B.2. Deep Learning Architectures

#### B.2.1. Iterative ADMM DL-based Accelerated MRI Reconstruction

A wide range of deep learning approaches have been proposed for accelerated MRI, with many relying on unrolled iterative schemes that embed the acquisition physics within a learned optimization procedure. Examples include gradient-descent unrolling in either image or frequency domains (Hammernik et al., 2017; Lønning et al., 2019; Sriram et al., 2020; Yiasemis et al., 2022) and first-order methods based on proximal gradient (Luo et al., 2023), conjugate gradient (Kim and Chung, 2022), or ADMM (Yiasemis et al., 2024a).

For our experiments in Accelerated MRI Reconstruction we employ a DL-based algorithm that exploits variable half-quadratic splitting followed by ADMM unrolled optimization spanning $J$ iterations, namely vSHARP (variable Splitting Half-quadratic ADMM algorithm for Reconstruction of inverse-Problems). Given undersampled $k$-space measurements $\tilde{\mathbf{y}}$, and sensitivity maps $\mathbf{S}$, each unrolled iteration comprises the following steps:

$$\mathbf{x}^{(j)} = \underset{\mathbf{x} \in \mathbb{C}^n}{\arg\min} \frac{1}{2} \sum_{k=1}^{n_c} \left\| \mathcal{A}_{\Theta,\mathbf{S}^k}^k(\mathbf{x}) - \tilde{\mathbf{y}}^k \right\|_2^2 + \frac{\eta_j}{2} \left\| \mathbf{x} - \mathbf{w}^{(j-1)} + \frac{\mathbf{u}^{(j-1)}}{\eta_j} \right\|_2^2, \tag{16a}$$

$$\mathbf{w}^{(j)} = \mathcal{D}_{\phi_j}(\mathbf{x}^{(j)}, \mathbf{w}^{(j-1)}, \frac{\mathbf{u}^{(j-1)}}{\eta_j}) \tag{16b}$$

$$\mathbf{u}^{(j)} = \mathbf{u}^{(j)} + \eta_j(\mathbf{x}^{(j)} - \mathbf{w}^{(j)}), \quad j = 1, \cdots, J. \tag{16c}$$

vSHARP solves (16a) via an iterative differentiable gradient scheme, while (16b) is learned using trainable convolutional-based denoising modules $\mathcal{D}_{\phi_j}$. Initial estimations of each variable is obtained as follows:

$$\mathbf{x}^{(0)} = \mathbf{z}^{(0)} := \sum_{k=1}^{n_c} \mathbf{S}_k^* \mathcal{F}^{-1}(\tilde{\mathbf{y}}), \quad \mathbf{u}^{(0)} = \mathcal{U}_{\phi_u}(\mathbf{x}^{(0)}), \tag{17}$$

where $\mathcal{U}_{\phi_u}$ represents a DL-based initializer comprising alternating sequences of dilated convolutions and replication padding responsible for predicting suitable initial value for the Lagrange Multiplier step in (16c). For further details refer to the original work (Yiasemis et al., 2024b).

For the prediction of the sensitivity maps $\mathbf{S}$, vSHARP also employs a separate DL convolutional-based model, denoted as $\mathcal{S}_{\phi_S}$, which takes as input estimated sensitivities $\tilde{\mathbf{S}}$ using ACS-sampled $k$-space data (see (Yiasemis et al., 2022) for more details on initial estimation) and refines them during training:

$$\mathbf{S}_k = \mathcal{S}_{\phi_S}(\tilde{\mathbf{S}}_k), \quad k = 1, \cdots, n_c. \tag{18}$$

Concerning the architecture of the denoising models $\{\mathcal{D}_{\phi_j}\}_{j=1}^J$ and sensitivity module $\mathcal{S}_{\phi_S}$, we opted for the 2D U-Net architecture (Ronneberger et al., 2015), which combines an encoder (2D convolutions and 2D max pooling), and a decoder (2D transpose convolutions) with skip connections.

### B.2.2. CBCT

For CBCT we use $\partial$U-net, a multi-scale learned iterative scheme that operates across four spatial resolutions (1, 1/2, 1/4, 1/8). The lowest resolution reconstruction is progressively refined through successive convolutional blocks, each consisting of three convolutional layers with ReLU activations and normalization layers. The final image is produced by a high-resolution 3D U-net that integrates the intermediate multi-scale estimates.

As initialization, we use a filtered backprojection (FDK) reconstruction with a ramp filter and a 95% frequency cut-off. All convolutional and transposed-convolutional layers in the backbone are replaced by modulated versions.

### B.2.3. FAN-BEAM

For Fan-beam CT we adopt the Learned Primal–Dual (LPD) algorithm, which unrolls the Primal–Dual Hybrid Gradient (PDHG) method. Each iteration consists of:

- a *primal update* in image space, implemented by a small CNN with three convolutional layers (PReLU activations and batch normalization),

- a *dual update* in projection space, parameterized by an analogous CNN,

with differentiable projection and backprojection operators linking the two domains. Because LPD is memory-intensive, it is applied only to the 2D fan-beam geometry.

All convolutional layers in both primal and dual modules are replaced by modulated convolutions.

## Appendix C. Experimental Setup

### C.1. Quantitative Evaluation Metrics

Let $x \in \mathbb{R}^{n_1 \times n_2}$ denote the reference image and $\hat{x} \in \mathbb{R}^{n_1 \times n_2}$ the reconstructed image. For convenience, let $N = n_1 \times n_2$ denote the total number of pixels.

**Structural Similarity Index Measure (SSIM)**

$$\mathrm{SSIM}(x, \hat{x}) = \frac{(2\mu_x \mu_{\hat{x}} + C_1)(2\sigma_{x\hat{x}} + C_2)}{(\mu_x^2 + \mu_{\hat{x}}^2 + C_1)(\sigma_x^2 + \sigma_{\hat{x}}^2 + C_2)}, \tag{19}$$

where $\mu_x, \mu_{\hat{x}}$ are local means; $\sigma_x^2, \sigma_{\hat{x}}^2$ local variances; $\sigma_{x\hat{x}}$ the local covariance; and $C_1, C_2$ stability constants.

**Peak Signal-to-Noise Ratio (pSNR)**

$$\mathrm{MSE}(x, \hat{x}) = \frac{1}{N} \sum_{i=1}^{N} (x_i - \hat{x}_i)^2, \tag{20}$$

$$\mathrm{pSNR}(x, \hat{x}) = 10 \log_{10} \left( \frac{(\max(x))^2}{\mathrm{MSE}(x, \hat{x})} \right). \tag{21}$$

**Normalized Mean Squared Error (NMSE)**

$$\mathrm{NMSE}(x, \hat{x}) = \frac{\|x - \hat{x}\|_2^2}{\|x\|_2^2}. \tag{22}$$

**Mean Absolute Error (MAE / L1)**

$$\mathrm{MAE}(x, \hat{x}) = \frac{1}{N} \sum_{i=1}^{N} |x_i - \hat{x}_i|. \tag{23}$$

Higher SSIM and pSNR values, alongside lower NMSE and MAE, indicate superior reconstruction fidelity.

### C.2. Triangular Distribution

During training the acceleration factor $R$ for accelerated MRI reconstruction and photon count $I_0$ for Computed Tomography were selected using a (right-angle) triangular distribution within an interval $[a, b]$ with peak at $b$. The idea of using this distribution is that, for instance in accelerated MRI Reconstruction, higher acceleration factors are generated more often motivated by the fact the reconstruction model sees less data for higher accelerations. The same idea can be transfered to Computed Tomography for $I_0$.

Below, we outline the definition of such distribution for arbitrary choices of $a$ and $b$. The triangular distribution in the range $[a, b]$ can be characterized by a Probability Density Function (PDF) that linearly increases from $a$ to $b$ as follows:

$$p(x) = \frac{2(x - a)}{(b - a)(b - a)} = \frac{2x}{b^2 - a^2}, \quad a \leq x \leq b. \tag{24}$$

The cumulative distribution function (CDF) and inverse CDF of $p$ are given by:

$$F(x) = \frac{x^2 - a^2}{b^2 - a^2}, \quad a \leq x \leq b, \tag{25}$$

$$\text{and } F^{-1}(u) = \sqrt{u \cdot (b^2 - a^2) + a^2}, \quad 0 \leq u \leq 1. \tag{26}$$

The inverse cdf $F^{-1}$ method can be applied to sample from $p$ as follows:

1. Generate a uniform random number $u' \sim U[0, 1]$.

2. Return $F^{-1}(u')$.

## C.3. Accelerated MRI Reconstruction

### C.3.1. TRAINING AND OPTIMIZATION DETAILS

**Model Optimization** All models were developed in PyTorch (Paszke et al., 2017) and optimized using Adam (Kingma and Ba, 2017) with parameters $(\beta_1, \beta_2) = (0.9, 0.999)$ and $\epsilon = 1\mathrm{e}{-8}$. Experiments were carried out on NVIDIA A100 or H100 GPUs. A batch size of 2 was used for static reconstruction and 1 for dynamic reconstruction. Static models were trained for 150,000 iterations, while dynamic models were trained for 80,000 iterations. The learning rate schedule began with an initial rate of 6.7e−4, increased linearly to 2e−3 over the first 1,000 iterations, and subsequently decayed by 20% every 30,000 iterations.

**Random Augmentations** During training across all setups, random augmentations were applied to improve model robustness and learning efficacy. These included random cropping ($320 \times 320$ regions for static reconstruction and $(n_t, n_x/3, n_y/2)$ random crops for dynamic reconstruction, where $n_t$, $n_x$, and $n_y$ denote the temporal and spatial dimensions), random horizontal or vertical flipping, and random rotation.

**Reconstruction Model Hyperparameters and Loss Function** The vSHARP models incorporated U-Nets with four scales for both denoising and sensitivity estimation. For denoising, 2D U-Nets with 32, 64, 128, and 256 filters were used for static reconstruction, while 3D U-Nets with 16, 32, 64, and 128 filters were used for dynamic reconstruction. Sensitivity estimation employed U-Nets with 16, 32, 64, and 128 channels. For static reconstruction, 12 denoising steps and 10 data consistency steps were used, while dynamic reconstruction employed 8 denoising steps and 6 data consistency steps. The Lagrange Multiplier module was in line with the original vSHARP framework (Yiasemis et al., 2024b). All remaining architectural and training choices were kept consistent with the experiments presented in (Yiasemis et al., 2024b). A dual-domain loss combining image-domain and $k$-space-domain loss components was employed, following the original work.

### C.3.2. COMPARISON AND ABLATION STUDIES

**Additional Results** We show the SSIM and pSNR results of our experiments for high acceleration factors in S1. Moreover, in Tab. S2 are provided the quantitative results for the NMSE metric for our experiments in Accelerated MRI Reconstruction in Sec. 4.3, corresponding to Tab. 1.

| | **Prostate Dataset** | | | | | |
|---|---|---|---|---|---|---|
| | Acceleration Factor ($R$) / ACS fraction ($r_{\text{acs}}$) | | | | | |
| **Method** | 12 / 0.03 | | 14 / 0.025 | | 16 / 0.02 | |
| | SSIM | pSNR | SSIM | pSNR | SSIM | pSNR |
| No MOD | 0.8104 | 31.44 | 0.7849 | 30.43 | 0.7648 | 29.47 |
| MOD S | 0.8086 | 31.32 | 0.7883 | 30.52 | 0.7677 | 29.72 |
| MOD M | 0.8097 | 31.42 | 0.7906 | 30.63 | 0.7720 | 29.88 |
| MOD L | **0.8151** | **31.63** | **0.7917** | **30.73** | **0.7745** | **29.93** |
| | **Knee Dataset** | | | | | |
| | Acceleration Factor ($R$) / ACS fraction ($r_{\text{acs}}$) | | | | | |
| **Method** | 12 / 0.03 | | 14 / 0.025 | | 16 / 0.02 | |
| | SSIM | pSNR | SSIM | pSNR | SSIM | pSNR |
| No MOD | 0.8523 | 33.62 | 0.8368 | 32.24 | 0.8172 | 30.46 |
| AdaIn | 0.8494 | 33.60 | 0.8333 | 32.23 | 0.8170 | 30.64 |
| MOD S | 0.8537 | 33.97 | 0.8412 | 32.70 | 0.8246 | 30.98 |
| MOD M (inp-only) | 0.8549 | 33.97 | 0.8422 | 32.77 | 0.8272 | 31.07 |
| MOD M | **0.8586** | **34.17** | **0.8432** | 32.64 | 0.8246 | 30.78 |
| MOD L | 0.8543 | 34.06 | 0.8412 | **32.80** | **0.8255** | **31.24** |

Table S1: Quantitative results (SSIM, pSNR) for Accelerated MRI Reconstruction at high accelerations.

| | **Prostate Dataset** | | | | | | |
|---|---|---|---|---|---|---|---|
| **Method** | Acceleration Factor ($R$) / ACS fraction ($r_{\text{acs}}$) | | | | | | |
| | 4 / 0.08 | 6 /0.06 | 8 / 0.04 | 10 / 0.035 | 12 / 0.03 | 14 / 0.025 | 16 / 0.02 |
| No MOD | 0.0058 | 0.0090 | 0.0121 | 0.0154 | 0.0186 | 0.0234 | 0.0290 |
| MOD S | 0.0053 | 0.0087 | 0.0124 | 0.0157 | 0.0191 | 0.0228 | 0.0274 |
| MOD M | **0.0053** | **0.0087** | 0.0123 | 0.0154 | 0.0186 | 0.0223 | 0.0265 |
| MOD L | 0.0053 | 0.0088 | **0.0118** | **0.0151** | **0.0178** | **0.0219** | **0.0260** |
| | **Knee Dataset** | | | | | | |
| **Method** | Acceleration Factor ($R$) / ACS fraction ($r_{\text{acs}}$) | | | | | | |
| | 4 / 0.08 | 6 /0.06 | 8 / 0.04 | 10 / 0.035 | 12 / 0.03 | 14 / 0.025 | 16 / 0.02 |
| No MOD | 0.0067 | 0.0083 | 0.0106 | 0.0134 | 0.0166 | 0.0228 | 0.0350 |
| AdaIn | 0.0064 | 0.0085 | 0.0109 | 0.0133 | 0.0166 | 0.0229 | 0.0334 |
| MOD S | 0.0069 | 0.0088 | 0.0111 | 0.0135 | 0.0158 | 0.0207 | 0.0315 |
| MOD M (inp-only) | 0.0066 | 0.0083 | 0.0106 | 0.0128 | 0.0154 | 0.0204 | 0.0301 |
| MOD M | **0.0061** | **0.0079** | **0.0100** | **0.0123** | **0.0146** | 0.0209 | 0.0334 |
| MOD L | 0.0063 | 0.0081 | 0.0104 | 0.0129 | 0.0154 | **0.0203** | **0.0295** |

Table S2: Quantitative results (NMSE) for Accelerated MRI Reconstruction.

**Additional Experiments**   To further assess the flexibility of the generalized modulated convolution, we conducted additional comparisons beyond the configurations evaluated in Sec. 4.3. Among the feature-modulation variants (MOD S, MOD M, MOD L), the best-performing configuration-referred to here as Feat (Best)-serves as our reference model. This variant corresponds to the strongest-performing feature-modulation setup in Tab. 1 and

Tab. S2, where modulation predicts a $C_\text{in} \times C_\text{out}$ feature-wise scaling matrix as defined in Eq. (6).

1. *Partial-in* modulation using one hidden layer with 32 input and 32 output features (Part-In L).

2. *Full* modulation using one hidden layer with 32 input and 8 output features (Full S).

| **Prostate Dataset** | | | | | | | | | | | | | | | | | | | | | |
|---|---|---|---|---|---|---|---|---|---|---|---|---|---|---|---|---|---|---|---|---|---|
| Acceleration Factor ($R$) / ACS fraction ($r_\text{acs}$) | | | | | | | | | | | | | | | | | | | | | |
| **Method** | 4 / 0.08 | | | 6 /0.06 | | | 8 / 0.04 | | | 10 / 0.035 | | | 12 / 0.03 | | | 14 / 0.025 | | | 16 / 0.02 | | |
| | SSIM | pSNR | NMSE | SSIM | pSNR | NMSE | SSIM | pSNR | NMSE | SSIM | pSNR | NMSE | SSIM | pSNR | NMSE | SSIM | pSNR | NMSE | SSIM | pSNR | NMSE |
| Feat (Best) | **0.9249** | **37.01** | **0.0053** | 0.8871 | 34.77 | 0.0087 | **0.8610** | **33.40** | **0.0118** | **0.8352** | **32.36** | **0.0151** | **0.8151** | **31.63** | **0.0178** | **0.7917** | **30.73** | **0.0219** | **0.7745** | **29.93** | **0.0260** |
| Part-In L | 0.9214 | 36.83 | 0.0055 | 0.8829 | 34.63 | 0.0089 | 0.8507 | 33.06 | 0.0128 | 0.8248 | 32.00 | 0.0163 | 0.8012 | 31.13 | 0.0199 | 0.7793 | 30.24 | 0.0244 | 0.7575 | 29.36 | 0.0298 |
| Full S | 0.9246 | 36.97 | 0.0053 | **0.8874** | **34.81** | **0.0086** | 0.8567 | 33.28 | 0.0122 | 0.8319 | 32.24 | 0.0155 | 0.8096 | 31.38 | 0.0188 | 0.7886 | 30.56 | 0.0227 | 0.7688 | 29.77 | 0.0272 |
| **Knee Dataset** | | | | | | | | | | | | | | | | | | | | | |
| Acceleration Factor ($R$) / ACS fraction ($r_\text{acs}$) | | | | | | | | | | | | | | | | | | | | | |
| **Method** | 4 / 0.08 | | | 6 /0.06 | | | 8 / 0.04 | | | 10 / 0.035 | | | 12 / 0.03 | | | 14 / 0.025 | | | 16 / 0.02 | | |
| | SSIM | pSNR | NMSE | SSIM | pSNR | NMSE | SSIM | pSNR | NMSE | SSIM | pSNR | NMSE | SSIM | pSNR | NMSE | SSIM | pSNR | NMSE | SSIM | pSNR | NMSE |
| Feat (Best) | 0.9096 | 38.71 | 0.0061 | **0.8926** | 37.22 | **0.0079** | **0.8802** | **36.06** | **0.0100** | **0.8679** | **35.05** | **0.0123** | **0.8586** | **34.17** | **0.0146** | **0.8432** | 32.80 | 0.0203 | 0.8255 | 31.24 | 0.0295 |
| Part-In L | 0.9085 | 38.72 | 0.0061 | 0.8913 | **37.27** | 0.0080 | 0.8780 | 36.02 | 0.0102 | 0.8654 | 35.01 | 0.0125 | 0.8556 | 34.14 | 0.0149 | 0.8424 | **32.88** | **0.0197** | **0.8265** | **32.32** | **0.0287** |
| Full S | **0.9097** | **38.79** | **0.0061** | 0.8918 | 37.20 | 0.0082 | 0.8738 | 35.93 | 0.0105 | 0.8649 | 34.82 | 0.0131 | 0.8545 | 33.87 | 0.0158 | 0.8405 | 32.54 | 0.0214 | 0.8226 | 30.84 | 0.0328 |

Table S3: Quantitative results for the additional experiments evaluating generalized modulation strategies on the prostate and knee MRI datasets. "Feat (Best)" denotes the strongest-performing feature-modulation variant selected from the MOD S/M/L configurations in Sec. 4.3, where modulation predicts a ($C_\text{out} \times C_\text{in}$) feature-scaling matrix. "Part-In L" corresponds to partial-in modulation using one hidden layer with 32 input and 32 output features, and "Full S" denotes a fully modulated variant with one hidden layer and 32 input and 8 output features.

The results in Tab. S3 indicate that both generalized variants perform competitively with Feat (Best) across all acceleration factors for both prostate and knee datasets. While Feat (Best) remains the strongest overall, the performance differences are small. This suggests that the primary benefit arises from including acquisition-aware conditioning itself, whereas the exact parametrization of the modulator has a secondary effect.

Table S4: Model size and inference time for 2D MRI reconstruction and 2D dynamic MRI. Inference time is reported per reconstructed volume.

| 2D MRI Reconstruction | | |
|---|---|---|
| Method | # Params (M) | Time (s) |
| No MOD | 95.10 | 10.9669 |
| AdaIN | 96.22 | 12.6548 |
| MOD S | 199.28 | 11.9411 |
| MOD M (inp conv) | 95.41 | 11.1739 |
| MOD M | 291.85 | 11.9791 |
| MOD L | 477.00 | 11.7859 |
| Partial Input | 109.08 | 11.7251 |

| 2D Dynamic MRI | | |
|---|---|---|
| Method | # Params (M) | Time (s) |
| No MOD | 47.30 | 5.5406 |
| MOD S | 252.23 | 5.7026 |

## C.4. Computed Tomography

**Additional Results**  In Tab. S5 are provided the quantitative results for the Fan-beam CT experiments. In Fig. S1 and Fig. S2 samples from variable projection count CBCT are provided.

| | Thorax Dataset | | | | | | | | | |
|---|---|---|---|---|---|---|---|---|---|---|
| Method | $I_0 = 2.5$k | | $I_0 = 5$k | | $I_0 = 10$k | | $I_0 = 20$k | | $I_0 = 40$k | |
| | MAE | pSNR | MAE | pSNR | MAE | pSNR | MAE | pSNR | MAE | pSNR |
| No MOD | 42.01 | 32.82 | 40.19 | 33.47 | 38.77 | 33.99 | 37.64 | 34.40 | 36.75 | 34.70 |
| MOD S | 42.12 | 32.80 | 40.22 | 33.45 | 38.75 | 33.99 | 37.58 | 34.41 | 36.67 | **34.74** |
| MOD M | 42.03 | 32.81 | **40.16** | 33.46 | **38.71** | 33.99 | **37.54** | 34.41 | **36.63** | 34.73 |
| MOD L | **42.01** | **32.82** | 40.17 | **33.47** | 38.73 | **34.00** | 37.58 | **34.41** | 36.68 | 34.73 |

Table S5: Quantitative results (MAE and pSNR) for Fan-beam CT experiments.

Table S6: Model size and inference time for 3D CBCT. Inference time is reported per reconstructed volume.

| 3D CBCT Reconstruction | | |
|---|---|---|
| Method | # Params (M) | Time (s) |
| No MOD | 24.66 | 1.76 |
| MOD S | 36.83 | 1.77 |

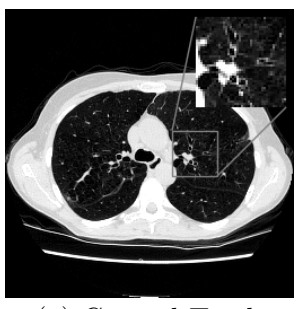 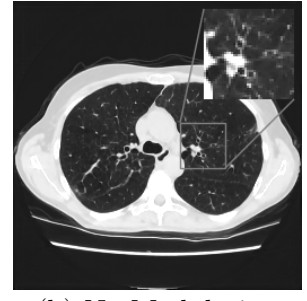 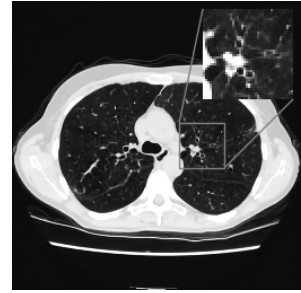

(a) Ground Truth · (b) No Modulation · (c) With Modulation

Figure S1: Axial slice of thorax CT for variable projection count CBCT experiment, 720 projections, HU range $[-1350, 150]$ and $[-1350, 0]$ for the ROI.

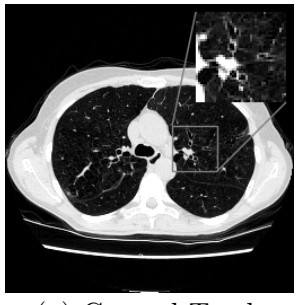 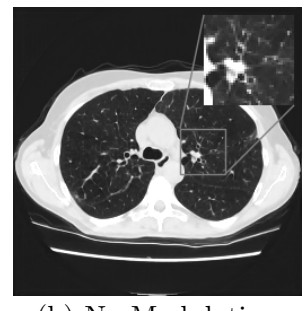 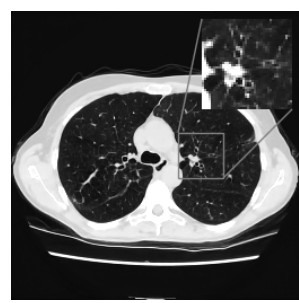

(a) Ground Truth · (b) No Modulation · (c) With Modulation

Figure S2: Axial slice of thorax CT for variable projection count CBCT experiment, 237 projections, HU range $[-1350, 150]$ and $[-1350, 0]$ for the ROI.

## Appendix D. Discussion

The experimental results across accelerated MRI, Cone-beam CT, and Fan-beam CT collectively underline the central observation of this work: conditioning learned iterative reconstruction schemes on acquisition parameters offers gains in reconstruction quality. The improvements emerge consistently across modalities, architectures, and parameter ranges, supporting the broader claim that variability in acquisition settings-when left implicit-limits the representational efficiency of non-conditional networks.

A recurring finding is that the magnitude of improvement depends strongly on the interaction between the physical forward model, the available measurement information, and the architecture used to perform the iterative updates. In accelerated MRI, where signal-to-noise characteristics and aliasing structure vary substantially with acceleration factor and ACS ratio, the modulated variants consistently outperform the unmodulated baselines across nearly all tested conditions. This is visible in Tab. 1 and Tab. S2, where modulated models demonstrate higher SSIM and pSNR, and lower NMSE, particularly at higher acceleration factors-precisely the regime where the underdetermined nature of the inverse problem becomes most severe. For the prostate data, the MOD L configuration provides the most stable gains, unlike the knee data, were MOD M was the best performer overall.

Furthermore, in cases of varying field strength in MRI, conditioning becomes particularly relevant, as field strength directly affects signal-to-noise characteristics and measurement statistics. In our dynamic experiments, using the Cardiac MRI Reconstruction Challenge 2025 dataset with 1.5T and 3T acquisitions, the modulated model consistently outperform the unmodulated baseline (Tab. 2), with larger gains observed when field strength is explicitly provided as conditioning information.

In Cone-beam CT, where the conditioning variable is the photon count $I_0$, improvements follow a similar trend. Tab. 3 demonstrates that lower-dose settings ($I_0 = 10k$) benefit most from modulation, with the modulated $\partial$U-net producing sharper soft-tissue detail (Fig. 5) and lower mean absolute errors. In Cone-beam CT with variable projection count where the projection count is the conditioning variable, improvements from conditioning are more pronounced as shown in Tab. 4 and reach 0.8 dB in PSNR. This is accomplished at negligible inferece time costs, as seen from Tab. S6.

Fan-beam CT results, however, show that the gains are smaller and sometimes marginal (Tab. S5). One plausible explanation, supported directly by the manuscript, is that LPD's dual blocks operate directly on the projection data. This allows the unmodulated network to infer noise characteristics implicitly. Consequently, external conditioning provides less additional value. Although the modulated variants still tend to outperform the baseline on average, the improvements are modest compared to MRI or CBCT.

A broader limitation of the present study is that only a subset of potentially relevant acquisition parameters was explored. For MRI, the auxiliary variable was restricted to acceleration factor, ACS ratio and field strength, but trajectory type, sequence type, or number of coils, are also meaningful candidates. Similarly, in CT, conditioning was performed on photon count and projection count in case of CBCT only. Exploring variables such as tube voltage may further clarify when and how conditioning is most impactful.

Another consideration is the trade-off between introducing modulation versus simply increasing model capacity. A wider U-Net or deeper CNN might recover part of the same performance gap by learning a richer set of shared filters. While exhaustively comparing these alternatives is beyond the scope of this work, the MRI results suggest that the gains from conditioning persist even when modulation capacity is relatively small (e.g., MOD S), indicating that explicit acquisition-aware adaptation cannot be trivially replaced by larger generic models.

Importantly, this distinction is further supported by the input-only modulation experiments. Modulating only the input convolution layers yields consistent improvements over the non-modulated baseline while introducing only a negligible increase in parameter count. This suggests that early, acquisition-aware adaptation of feature extraction already captures a substantial fraction of the benefit of conditioning, without requiring full modulation throughout the network. In this sense, input-level modulation acts as a lightweight yet effective mechanism for aligning the reconstruction process with acquisition-specific statistics, reinforcing that the observed gains stem from conditioning rather than from increased representational capacity alone.

It is also worth noting that our experiments rely on three specific learned iterative schemes, each selected as a representative architecture for its respective inverse problem: an ADMM-based unrolled model for MRI, $\partial$U-Net for CBCT, and Learned Primal–Dual for fan-beam CT. While many other reconstruction backbones exist, we did not attempt

to evaluate modulation across the full architectural spectrum. Nevertheless, these choices comprise representative designs within their modalities, and therefore provide a reasonable basis for assessing how acquisition-aware conditioning behaves in practice.

In addition, the generalized formulation in Appendix B.1 shows that several modulation parameterizations are possible. Under the configurations evaluated here for MRI (see Tab. S3), our original feature-wise (channel-wise) formulation performs best overall and does so with the lowest computational overhead. It is also worth noting that the comparisons are not entirely fair: the Partial-In variant was tested with a larger MLP, whereas the Full variant used a smaller one, which makes their relative performance harder to interpret directly.

The triangular sampling strategy used during training as described in Appendix C.2 emphasizes challenging regimes (high $R$ in MRI, low $I_0$ in CT). This likely contributes to the larger gains observed in these regions. Whether a uniform or different sampling schedule would yield different patterns-particularly at lower accelerations-remains an open question.

Lastly, beyond the specific feature-wise modulation strategy evaluated in this work, several alternative conditioning mechanisms exist. A widely used family of approaches relies on feature-wise affine transformations, such as FiLM layers (Perez et al., 2018; Dumoulin et al., 2018), which scale and shift intermediate activations based on auxiliary variables. A different line of methods introduces conditional sparsity through learnable gating or $L_0$-regularization (Louizos et al., 2017; He et al., 2017), effectively modulating the set of active channels or filters in response to the input. These strategies demonstrate that modulation can be implemented in multiple architectural forms, ranging from explicit feature-wise transformations to implicit capacity control. Comparing such alternatives with the proposed design is a natural extension for future work.

Related work has also introduced adaptive mechanisms within MRI-specific reconstruction frameworks. Examples include approaches that adapt the reconstruction model to different acquisition settings (Pramanik et al., 2023) and hypernetwork-based methods that generate parts of the reconstruction network from coil- or scanner-specific embeddings (Ramanarayanan et al., 2023a), as well as recent adaptive convolution approaches for QSM dipole inversion, where convolution kernels are generated as functions of acquisition geometry within a feed-forward U-Net architecture (Graf et al., 2024).

These methods demonstrate that acquisition-aware adaptiveness is feasible in MRI reconstruction, but differ in scope and mechanism from the present work, which introduces lightweight modulation of convolutional operators within learned iterative reconstruction schemes.

Overall, the results indicate that conditional learned iterative schemes offer a simple, architecture-agnostic mechanism for adapting deep reconstruction models to heterogeneous acquisition settings without training separate networks for each configuration. The improvements are consistent, and their strength depends on both the modality and the reconstruction backbone. Expanding the conditioning variables, exploring interactions with model capacity, and evaluating additional architectures represent natural next steps for future work.

