# OpenReview forum: "Conditional Learned Reconstruction for Medical Imaging"
_MIDL.io/2026/Conference — MIDL 2026 Poster_

### Official Review · Reviewer_XGGQ · 2026-01-09

**Confidence:** 4
**Preliminary Rating:** 3
**Final Rating:** 4

**Summary:**

This paper explores the use of a specific filter generating network, termed as conditionally learned reconstruction, in the context of medical imaging reconstruction. Application is demonstrated on MRI and CT image reconstruction tasks and the authors show modest improvement in performance with their approach

**Strengths:**

Well written and easy to follow paper dealing with clinically relevant problems in CT and MRI. The use of modulated convolutions, while not entirely novel, is an area that deserves further exploration. Medical imaging systems and that imagery they generate are incredibly complex and innovations that leverage additional information to improve imaging tasks are of great value.

**Weaknesses:**

See comments below. See comments below. See comments below. See comments below. See comments below. See comments below. See comments below. See comments below. See comments below. See comments below. See comments below.

**Detailed Comments:**

Filter generating networks have been around for several decades now in some form or the other. While their use in medical image reconstruction / processing is still relatively new, its worth adding some detail in the introduction. Currently, I only see Hypernetwork based methods in your discussion. Recommend expanding that discussion (and include some context in the early sections...maybe the Introduction)

Can recommend adding the following work to your discussion as a related idea: Incorporating a-priori information in deep learning models for quantitative susceptibility mapping via adaptive convolution by Graf et al. While dealing with a different application, the core idea of adapting / modulating convolutions has been explored. I would urge the authors to do a more rigorous literature survey and update the discussion so that their work can be better characterized along side existing literature

I dont understand the rational behind the specific conditioning variables used in this work. In the MR case, the acceleration rate and ACS fraction are used - this is essentially information that a network can glean from the measured data trivially. MRI provides a rich source of attributes to use: contrast information; noise standard deviation etc., which one would expect to be more suitable for conditioning. Similarly, on the CT side, the use of the background scan (I0) term for conditioning instead of something like the tube current etc., needs additional clarification. Please expand upon these

While I understand the overall improvements are modest, I am concerned with the choice of images used in Figure 3. 2D MRI is a challenging setting for high acceleration and in this reviewers opinion R = 4 - 6 is likely the limit of what today's deep learning technology can accomplish. Your R=16 image in Figure 3 is not going be clinically meaningful - would recommend updating it to a lower acceleration and show the fidelity with ground truth

The MRI experiments in particular are likely valuable in the R = 2- 6 range. If possible, please consider updating your experiments to this range.

**Justification Of Final Rating:**

The authors have done a reasonable job updating the manuscript and I am happy to recommend acceptance of this work. I expect it to generate interesting discussion with audience interested in image reconstruction

**Justification Of The Preliminary Rating:**

Paper is well written and explores a potentially useful idea that can enhance current medical imaging reconstruction algorithms. I expect the authors to be able to update the draft readily to account for my feedback after which I can recommend acceptance

**Questions To Address In The Rebuttal:**

Most of my comments can likely be addressed with updates to discussion / introduction sections.

I suggest considering R = 2- 6 acceleration range for MRI to better explore performance in a regimen where the generated imagery can likely have some clinical value

---

> ### Author Response · Authors · 2026-01-25
> **Responses to Reviewer  XGGQ**
>
> We thank the reviewer for the thoughtful and constructive feedback, and for recognizing the clinical relevance of the problem setting. Below we address each point in detail.
>
> ---
>
> ### 1. Relation to prior work on filter-generating and adaptive convolution methods
>
> We thank the reviewer for the suggestion to expand and clarify the positioning of our work within the literature on conditioning and adaptive convolution methods.
>
> In response, we have expanded the Introduction (Section 1) to explicitly discuss prior conditioning strategies, including feature-wise modulation (e.g., FiLM [1] and Adaptive Instance Normalization [2]), hypernetwork-based approaches for reconstruction, and recent adaptive convolution methods such as Graf et al. for QSM [3]. This addition clarifies that conditioning in the AI field via auxiliary information has been explored in different forms and application domains.
>
> We further clarify how our approach differs from these lines of work. Rather than conditioning feature activations or generating a full set of reconstruction weights, we propose partial modulation of convolutional kernels and biases inside learned iterative reconstruction schemes. This design preserves the structure of the underlying optimization algorithm and forward model, while enabling sample-specific adaptation to acquisition parameters.
>
> We believe this expanded discussion more accurately positions our contribution relative to existing filter-generating and adaptive convolution approaches, and clarifies that the novelty lies in integrating lightweight, acquisition-aware operator modulation into learned iterative reconstruction frameworks and evaluating it systematically across MRI and CT.
>
> ---
>
> ### 2. Choice of conditioning variables in MRI and CT
>
> We appreciate the reviewer’s request for further clarification on the rationale behind the selected conditioning variables. MRI (acceleration factor and ACS fraction). Acceleration factor and ACS fraction directly determine the sampling pattern, data availability, and noise amplification in accelerated MRI, and therefore strongly influence the optimal regularization strength in iterative reconstruction. While it is true that a sufficiently expressive network could, in principle, infer these quantities implicitly from the measured data, our goal is to explicitly provide this information to avoid forcing the network to disentangle acquisition effects from anatomy. Importantly, these parameters are always known at acquisition time.
>
> To further address this point, we have added new dynamic cardiac MRI experiments in which field strength (1.5T vs. 3T) is included as an additional conditioning variable (see Additional experiments comment below). These results demonstrate that when richer acquisition metadata are available, the proposed framework naturally accommodates them and yields improvements.
>
> In CT, tube current affects image quality primarily through its impact on the photon count, which directly governs the noise statistics in the Poisson acquisition model. Since our simulator explicitly models photon count, conditioning on photon count provides a direct and physically meaningful representation of tube current effects, without requiring additional assumptions about scanner-specific calibration.
>
> ---
>
> ### 3. Clinical relevance of MRI acceleration factors and qualitative figures
>
> We agree that extremely high accelerations are not always clinically meaningful in 2D MRI. To address this concern, we have updated the qualitative results accordingly. Specifically, the knee MRI example previously shown at R=16 has been replaced by a lower acceleration setting (R=6), where reconstructions are more representative of clinically relevant regimes.
>
> We additionally include qualitative results for the cardiac MRI experiments (R=4, Figure 4).
>
> Regarding quantitative evaluation, while higher acceleration factors were included to stress-test the conditioning mechanism, the main paper now emphasizes the R=4-10 regime (Table 1), with higher accelerations moved to the Appendix (Table S1).
>
> ---

---

> ### Author Response · Authors · 2026-01-25
> **Responses to Reviewer XGGQ (continued)**
>
> ### 4. Acceleration range R=2-6
>
> We agree that the R=2-6 range is particularly relevant for many clinical MRI applications. In our experiments, R=4-6 already lies within this regime and is now more prominently highlighted in both quantitative and qualitative evaluations.
>
> Extending the study to include R=2 would unfortunately require retraining and rebalancing the sampling distribution, as our training strategy intentionally emphasizes higher accelerations to expose the model to more challenging conditions.
>
>
> ---
>
> ### 5. Additional experiments and conditioning complexity
>
> In response to concerns about conditioning complexity, we added a lightweight modulation variant in which conditioning is applied only at the first convolutional scale of the U-Net denoiser (MOD M - input-only, fastMRI knee). Despite its substantially reduced parameter count, this variant still consistently outperforms the non-modulated baseline, indicating that even minimal conditioning can be beneficial.
>
> We have also:
>
> - Extended the experiments to dynamic cardiac MRI (Table 2, Figure 4). This is due to the fact we now have data which include field strength [1] as an additional conditioning variable, demonstrating that the proposed framework naturally accommodates richer acquisition metadata when available.
> - Added experiments with variable projection count CBCT have been added (in addition to variable photon count). It is shown that conditioning on projection count increases PSNR by up to 0.8 dB PSNR (at negligible inference time cost).
>
> These additions demonstrate that the proposed framework generalizes across modalities, acquisition conditions, and reconstruction settings.
>
>
> ---
>
> ### 6. Computational cost
>
> We now explicitly report parameter counts and inference times for:
>
> - 2D accelerated MRI (fastMRI knee),
> - 2D+time dynamic cardiac MRI, and
> - 3D CBCT reconstruction.
>
> The results show that the proposed conditioning introduces modest memory overhead and negligible inference-time cost, supporting its practicality.
>
> ---
>
> ### 7. On tube voltage and current
>
> Since we didn’t incorporate the associated physics into our projection simulator, we removed reference to tube voltage in the abstract and the introduction to avoid potential overstatement. Tube current, however, directly affects the resulting photon count, which is why we condition on simulated photon count directly.
>
> ---
>
> We thank the reviewer again for the constructive feedback. We believe the expanded discussion, clarified rationale, updated qualitative results, and additional experiments directly address the raised concerns and strengthen the paper’s positioning. We hope these revisions clarify the contribution and support a positive recommendation.
>
> ### References
>
> [1] Ethan Perez, Florian Strub, Harm De Vries, Vincent Dumoulin, and Aaron Courville. Film: Visual reasoning with a general conditioning layer. In Proceedings of the AAAI conference on artificial intelligence, volume 32, 2018.
>
> [2] Xun Huang and Serge Belongie. Arbitrary style transfer in real-time with adaptive instance normalization. In Proceedings of the IEEE international conference on computer vision, pages 1501-1510, 2017.
>
> [3] Graf, Simon, Walter A. Wohlgemuth, and Andreas Deistung. "Incorporating a-priori information in deep learning models for quantitative susceptibility mapping via adaptive convolution." Frontiers in Neuroscience 18 (2024): 1366165.
>
> [4] Daryl Xu, "CMRxRecon2025", IEEE Dataport, February 25, 2025, doi:10.21227/b6xs-gv29

---

> ### Author Response · Authors · 2026-01-29
>
> We thank the reviewer again for the thoughtful and constructive feedback. We hope that our responses and revisions satisfactorily address the raised points, and we remain happy to provide any further clarification during the remainder of the discussion period.

---

### Official Review · Reviewer_dK8W · 2026-01-09

**Confidence:** 5
**Preliminary Rating:** 3
**Final Rating:** 3

**Summary:**

The authors introduce a framework where they update the neural network parameters based on the protocol-dependent acquisition parameters, along with the rawdata, for MRI and CT reconstruction. They formulate a technique called "modulated convolution", where they used a lightweight "modulator" network (MLP) to adjust convolutional kernels based on the specific scanner settings for a single sample. Experiments across MRI, CBCT, and FBCT show that these conditional schemes consistently outperformed non-modulated baselines, particularly in challenging high-noise or high-acceleration regimes.

**Strengths:**

1. Instead of any static conditioning, the authors use a "Modulator" (MLP) to adapt convolutional weights as a function of the scanner settings.
2. Validated across 2 different modalities data, MRI (Prostate, Knee) & CT (CBCT, FBCT)

**Weaknesses:**

1. In abstract, the authors mention about using "field strength" as one of the acquisition parameter to be utilized for a MRI recon architecture. But in their work, they don't use them. Although they mention it in their limitation section. Same case happened for CT (Tube voltage, Tube current).

2. The idea of “Modulating convolution weights using an MLP of scanner-specific conditioning variables” is conceptually close to hypernetworks / conditional conv. While the idea is pretty simple but elegant, current methods (ref1, ref2) use the acquisition parameters mostly as embeddings, not like the feature extractor used in the paper. But the paper misses this crucial comparison of using the parameters through a simple embedding (using much less parameters than a MLP). This work sets them as the key precedent to their work, yet missing this simple comparison.
They mentioned in the paper that "This suggests that the primary benefit arises from including acquisition-aware conditioning itself, whereas the exact parameterization of the modulator has a secondary effect." So it would be beneficial to have the acquisition-aware parameters having a very simple embedding with much less parameter count.

3. Prostate and knee datasets are both fastMRI - likely similar scanner/protocol characteristics. Whereas, recent works shows improvement using different scanner/ protocol charateristics (ref3). Real clinical variability includes field strength, coil configuration, vendor differences - but none tested in the work.

References:
1. Ref1: HyperCoil-Recon: A Hypernetwork-based Adaptive Coil Configuration Task Switching Network for MRI Reconstruction (Ramanarayanan et al., 2023)
2. Ref2: Adapting model-based deep learning to multiple acquisition conditions: Ada-MoDL (Pramanik et al., 2023)
3. Ref3: HierAdaptMR: Cross-Center Cardiac MRI Reconstruction with Hierarchical Feature Adapters (Xu et al. 2025)

**Detailed Comments:**

1. In abstract, the mention about "field strength" and "tube voltage, tube current" should be removed. As it creates a impression on the reader that the authors most likely used them in their work.

2. Keeping the acquisition-aware parameters as a simple embedding (i.e. getting rid of a overhead parameterized MLP) should be studied, as mentioned in the weakness.

3. A trade-off curve/ statistics between parameter count (i.e. memory overhead/ inference times) and modulation complexity should be explored.

**Justification Of Final Rating:**

The authors added parameter analyses, new cardiac MRI and CBCT experiments, and corrected overstated claims in the abstract. However, they did not compare their MLP modulator to simple embedding-based conditioning (as used in the cited works Ada-MoDL and HyperCoil-Recon), arguing that it requires architectural redesigns beyond their scope. Since the main concern was whether the modulator's added complexity is justified over simpler alternatives—especially given the authors admitted "acquisition-awareness matters more than modulator design"—this missing comparison prevents me from a score increase. The rebuttal shows the method works in more scenarios but doesn't prove it's better than cheaper alternative.

**Justification Of The Preliminary Rating:**

The paper addresses a relevant problem—adapting reconstruction networks to varying acquisition parameters—with a simple and intuitive approach. However, key weaknesses limit its contribution.

1. The abstract mentions field strength and tube voltage/current as conditioning variables, yet none are used experimentally. This overstates the scope.
2. Despite citing Ada-MoDL and HyperCoil-Recon as related work, the paper omits comparison to simpler embedding-based conditioning. The authors themselves note that acquisition-awareness matters more than modulator design, making this comparison essential.
4. Parameter counts and inference times for the modulator overhead are absent.

The idea is reasonable and results show consistent (if modest) improvements, but incomplete experiments and overclaimed scope prevent a stronger recommendation.

**Questions To Address In The Rebuttal:**

Please refer to the Detailed Comments section.

---

> ### Author Response · Authors · 2026-01-25
> **Responses to Reviewer dK8W**
>
> We thank the reviewer for the thorough and detailed assessment of our manuscript, as well as the concrete suggestions for strengthening the paper. Below, we address each point raised in detail.
>
> ---
>
> ### 1. Parameter costs and inference times
>
> We agree that quantifying the overhead introduced by modulation is important. We have therefore added a detailed analysis of parameter counts and inference times for the MRI and CBCT experiments. These are now reported in Table S4 (MRI) and Table S6 (CBCT). Across all evaluated settings, we observe that while larger modulation variants increase the number of learnable parameters, inference times remain comparable to the non-modulated baselines, confirming that the proposed conditioning introduces negligible runtime overhead. This directly addresses the concern regarding practical deployability.
>
> ---
>
> ### 2. Field strength and Tube Voltage/Current
>
> We agree with the reviewer that the original wording might have overstated the experimentally validated scope. Our goal was not to overstate the contributions, it was rather to show our rationale.
>
> #### **MRI (field strength).**
>
> While field strength was originally discussed as a motivating example, it was not evaluated in the initial version due to dataset limitations (fastMRI knee data include scans from both 1.5T and 3T scanners, but field strength is not provided in the released metadata). To address this, we have now added new experiments using the Cardiac MRI Reconstruction Challenge 2025 dataset [1], which explicitly includes both 1.5T and 3T acquisitions.
>
> We treat this as a 2D+time dynamic reconstruction problem and condition the model on acceleration factor, ACS fraction, and field strength. The results are now reported in Table 2 and Figure 4.
>
> Importantly, we observe that:
>
> - Modulated models consistently outperform non-modulated baselines.
> - The performance gap is more pronounced when field strength is included, supporting the central claim that explicit acquisition-aware conditioning is beneficial when such metadata are available.
>
> #### **CT (tube voltage and tube current).**
>
> We have removed references to tube voltage from the abstract and introduction. Tube voltage affects the X-ray energy spectrum and would require a more detailed spectral forward model, which is not incorporated in our projection simulator.
> Tube current, however, directly determines the photon count, which is explicitly modeled in our Poisson noise formulation. For this reason, we believe that conditioning on photon count is an appropriate and physically meaningful proxy for tube current, and we retain this terminology in the abstract.
>
> In addition, we have added a new CBCT experiment with variable projection count, reflecting real clinical variability (e.g., pelvic CBCT acquisitions without phase resolution). Quantitative results are provided in the main text (Table 4), with qualitative examples in the Appendix (Figures S1, S2). Conditioning on projection count yields up to 0.8 dB PSNR improvement, again at negligible inference-time cost (Table S6).
>
> ---

---

> ### Author Response · Authors · 2026-01-25
> **Responses to Reviewer dK8W (continued)**
>
> ### 3. Embedding-based conditioning vs. modulated convolutions]
>
> We thank the reviewer for this suggestion. We agree that comparing different parameterizations of acquisition-aware conditioning is a meaningful direction. However, we would like to clarify the scope and positioning of this work.
>
> Our goal is not to argue that modulated convolutions are categorically superior to all simpler conditioning mechanisms applied in the broader literature, but rather to demonstrate that:
>
> Explicit conditioning on acquisition parameters is beneficial across modalities (CT and MRI) and architectures (vSHARP, ∂U-Net, LPD) and extends beyond MRI-only experiments given in the cited literature.
> Modulated convolutions provide a lightweight, architecture-agnostic mechanism that integrates naturally into learned iterative schemes, without altering their optimization structure.
>
> Direct comparisons to embedding-based conditioning (e.g., injecting acquisition parameters as feature embeddings) would require non-trivial architectural redesigns for each backbone (vSHARP, ∂U-Net, LPD), potentially confounding the analysis with model-specific choices. For this reason, we consider such comparisons outside the intended scope of this paper.
>
> That said, we have taken steps to address the reviewer’s underlying concern about conditioning complexity versus benefit:
>
> We added a low-cost modulation variant where conditioning is applied only at the first (input) convolutional scale of the U-Net denoiser (MOD M - input-only, fastMRI knee). Despite introducing substantially fewer additional parameters, this variant still consistently outperforms the non-modulated baseline, demonstrating that even minimal conditioning can be effective.
> The parameter and runtime analysis (Tables S4 and S6) explicitly quantifies the trade-off between modulation complexity and overhead.
> The newly added field-strength-conditioned cardiac MRI experiments further demonstrate that the modulation framework naturally accommodates richer acquisition metadata when available.
>
> Together, these results support the reviewer’s point that acquisition awareness itself is the dominant factor, while also showing that the proposed modulation strategy can operate effectively even with low overhead.
>
> ---
>
> ### 4. Clinical variability and dataset diversity
>
> We agree that real-world clinical variability extends beyond the fastMRI knee and prostate datasets. This motivated the inclusion of:
>
> - Dynamic cardiac MRI with variable field strength, and
> - CBCT experiments with both variable photon count and variable projection count.
>
> These additions broaden the evaluation across acquisition conditions that more closely reflect clinical variability, while remaining grounded in datasets and forward models available to us.
>
> ---
>
> We hope that the added experiments, clarified scope, and explicit parameter/runtime analysis address the reviewer’s concerns. Importantly, all new results reinforce the central message of the paper: explicit, acquisition-aware conditioning improves learned reconstruction robustness across modalities, tasks, and acquisition regimes, even when implemented in lightweight forms.
>
>
> ---
>
> ### References
>
> [1] Daryl Xu, "CMRxRecon2025", IEEE Dataport, February 25, 2025, doi:10.21227/b6xs-gv29

---

> ### Author Response · Authors · 2026-01-29
>
> We thank the reviewer again for the careful and constructive feedback. We hope that our responses and revisions address the raised points satisfactorily, and we remain available for any further clarification during the remainder of the discussion period.

---

### Official Review · Reviewer_JMgf · 2026-01-11

**Confidence:** 5
**Preliminary Rating:** 3
**Final Rating:** 4

**Summary:**

This work introduces a framework for conditional learned iterative schemes in inverse imaging problems of CT and MR reconstruction, utilizing a modulation operation in the convolution step at the architecture level to adjust network parameters based on
physical acquisition settings. The authors compare conditional learned iterative schemes to their counterparts without conditioning
for both tomography and MRI to demonstrate their effectiveness.

**Strengths:**

The direction of using scan adaptive modulation in convolution for architectures used in iterative methods in medical imaging is welcoming and promising.

The authors have considered both CT and MRI which are key anatomical modalities in medical imaging.

The use of r_{acs} value to modulate convolution stands different from prior literature and seems a good contribution in this work. This also gives an impression that the work is strongly driven by the physiology of the scan system which an important problem to address.

Use of methods like Iterative ADMM-Net, LPD is very encouraging.

**Weaknesses:**

The second and third paragraphs in the introduction is a standard rote, the authors can make it shorter and explain more details about their contributions rather.

There have been feature modulation in several prior works in the same line of work [1] and [2] for data driven and iterative schemes. The authors are encouraged to refer them.

[1] Benefits of Linear Conditioning with Metadata for Image Segmentation, MIDL 2021

[2] MCI-HyperNet: A multiple contextual information-based adaptive weight learning network for controllable image reconstruction,
Neurocomputing, Volume 554, 2023

Although equation 6 explains the modulation process in Section 3.1. Modulated Convolution, this is the key contribution in their work and they have to add a figure to explain it. This improves paper quality and readability. Also the figure can explain how $f_\theta$ and $g_\psi$ are modeled

Figure 4 shows overly smoothed images. I believe it might be due to significantly less number of projections. If so, please mention them

The authors should compare their work against adaptive instance normalization applied to the baseline models. That way, the readers can understand the merits of their work.

**Detailed Comments:**

I don't have major questions to ask as the work is simple yet effective as the methods taken for experimentation of modulated convolution are physics-based.

**Justification Of Final Rating:**

I would like to improve my rating, as I appreciate the comparative study added to the manuscript and the observation agree with prior works. Also, adaptive methods for iterative schemes are still less explored than those for data-driven methods.

**Justification Of The Preliminary Rating:**

The authors consider very important methods in the classical deep learning literature. The modulated convolutions given in the literature are mostly data driven except for a few. The authors are encouraged to refer other feature modulation schemes like adaptive instance normalization in their work.

**Questions To Address In The Rebuttal:**

Are there any particular reasons for observing improvements in the variants - MOD S, M and L for different datasets? Does resolution play a role in this analysis?

The outputs of $f_\theta$ and $g_\psi$  are used for scaling and shifting which is similar to adaptive instance normalization. Please comment upon this.
Please take a look at this paper [3].

[3] Learning disentangled representations in the imaging domain,
Medical Image Analysis,
Volume 80,
2022

If the modulation in other methods are different then the authors should draw parallel on what is similar and dissimilar against [1] and [2], referring to [3].

what is . and * in equation 6?

---

> ### Author Response · Authors · 2026-01-25
> **Response to Reviewer JMgf**
>
> We thank the reviewer for the careful assessment of our manuscript and for the detailed and technically informed feedback. We appreciate the recognition of the relevance of scan-adaptive modulation for learned iterative reconstruction, as well as the constructive suggestions. Below, we address each point in detail.
>
> ---
>
> ### 1. Comparison to other methods
>
> In Lemay et al. the authors propose feature-wise linear modulation (FiLM) to condition a segmentation network on auxiliary metadata, where learned affine parameters scale and shift intermediate feature maps based on discrete side information such as tumor type or organ label. This conditioning operates after convolution and modulates activations rather than the convolutional operators themselves.
>
> In contrast, our work introduces modulated convolutions, where the auxiliary variables directly parameterize the convolutional kernels and biases through a learned function of the acquisition settings. Rather than post-hoc feature modulation, the conditioning acts at the level of the linear operator itself, effectively adapting the network’s filters to the physical acquisition regime of each sample.
>
> In Ramanarayanan et al. the authors employ a hypernetwork that generates or adapts the weights of a reconstruction network based on contextual information, effectively learning a mapping from acquisition parameters to model parameters. This design yields a context-specific reconstruction model whose weights change as a function of the acquisition setting.
>
> MCI-HyperNet differs from our approach in how conditioning is realized. We do not use a separate hypernetwork to generate a full set of reconstruction weights; instead, we introduce a modulation mechanism inside the reconstruction backbone by conditioning the convolutional kernels (and biases) directly via the acquisition variables.
>
>
> In Liu et al. (2022), the authors study disentangled representation learning and review conditioning mechanisms such as AdaIN, FiLM, and SPADE, where auxiliary variables modulate feature activations or normalization statistics to separate content and style or to encourage invariance to nuisance factors. Conditioning is applied post-convolution and primarily serves representation disentanglement and domain transfer, rather than explicitly adapting the reconstruction operator itself.
>
> In our work, we instead propose modulated convolutions, where the convolutional kernels themselves are conditioned on physical acquisition parameters (e.g., acceleration factor, ACS fraction, tube current). This induces sample-specific weight adaptation inside learned iterative reconstruction schemes, directly linking acquisition physics to the learned regularizer. Unlike feature-wise modulation, our approach alters the effective reconstruction operator and regularization strength in a principled way, without relying on latent disentanglement assumptions or post-hoc normalization layers.
>
> We have added a short discussion in the revised manuscript paper in Section 1 (Introduction).
>
> ---
>
> ### 2. Adaptive Instance Normalization and relation to f and g MLPs
>
> Adaptive Instance Normalization (AdaIN) method modulates intermediate feature activations by adjusting per-channel statistics (mean and variance) after the convolution is applied:
>
> x = conv(x)
> x = adain(x, z) # z is the auxiliary variable
> adain(x, z) = (x - x_mean) / sqrt(var) *  f1(z) + f2(z),
>
> where f1 and f2 act as weight and bias on the channel dimensions after normalization has been applied.
>
> The conditioning therefore operates on normalized activations and does not alter the underlying convolutional operator itself. As discussed in Liu et al. [3], this design is well suited for style transfer, domain adaptation, or representation disentanglement, where the goal is to separate content from nuisance variation.
>
> In contrast, in our formulation f_\psi and g_\psi aim to ​ parameterize (modulate) the convolutional weights and biases directly, before the convolution is applied.
>
> x = mod_conv(x, z)
>
> where the weight and bias of the convolution are modulated by z though f_\psi and g_\psi.
>
> The modulation therefore acts on the linear operator defining the learned regularizer inside each iteration of the reconstruction scheme. As a result, conditioning changes the effective filter response itself rather than rescaling already-computed feature maps. This distinction is particularly relevant in learned iterative reconstruction, where convolutional blocks play the role of learned proximal or denoising operators and directly control the strength and structure of regularization. We have clarified this distinction explicitly in the revised manuscript and added references to AdaIN-style conditioning to better situate our approach within the broader conditioning literature.
>
> While both the AdaIn and our modulator networks play an analogous role to scaling and shifting, they are fundamentally different in what they modulate.

---

> ### Author Response · Authors · 2026-01-25
> **Response to Reviewer JMgf (continued)**
>
> ### 3. Modulation Variants
>
> The MOD variants differ in the capacity of the modulation networks. In the accelerated MRI experiments, no single variant dominates across datasets. On the prostate dataset (Table 1, Tables S1-S2), higher-capacity modulation (MOD L) consistently improves performance, while smaller variants are sometimes comparable to or worse than the non-modulated baseline. In contrast, on the knee dataset (Table 1), a mid-capacity modulator (MOD M) performs best on average, and further increasing modulation capacity does not yield consistent gains. This suggests a trade-off between modulation expressivity and constraint rather than a universally optimal modulation size. We do not isolate resolution or anatomy as independent causal factors and therefore refrain from attributing the observed behavior to a single dataset property.
>
> ---
>
> ### 4. Comparison to Adaptive Instance Normalization
>
> We agree with the reviewer that a comparison to AdaIN-based conditioning is valuable. Since the U-Net denoisers used in our accelerated MRI experiments already employ instance normalization, we replaced standard instance normalization with adaptive instance normalization conditioned on the same auxiliary variables and evaluated this variant on the knee dataset (AdaIn M - since we use 32, 16 feature MLPs). We report these results alongside our proposed modulation strategy. This comparison allows us to disentangle the effect of conditioning via normalization statistics from conditioning the convolutional operators themselves. (These experiments are included in the revised manuscript, Table 1).
>
> We observe that adaptive instance normalization performs similarly to, or slightly worse than, the non-modulated baseline, and does not achieve the consistent improvements obtained with modulated convolutions. This suggests that, in this setting, conditioning through normalization statistics alone is less effective than directly modulating the convolutional operators with acquisition parameters.
>
> ---
>
> ### 5. Clarification of operators in Equation (6)
>
> In Equation (6), the symbol “.” denotes regular multiplication (of scalars) between the modulation weights W_\theta_m_k ​ and the base convolutional kernel weights 𝑘_m_k, while “*” denotes the standard cross-correlation (convolution) operation between the modulated kernel and the input feature maps. Note that we follow the convolution notation as in PyTorch [4].  We have clarified this explicitly in the text to avoid ambiguity.
>
> ---
>
> ### 6. Modulated Convolution Figure and f,g modelling
>
> We thank the reviewer for this suggestion. We have revised Figures 1 and 2 to more clearly illustrate the modulated convolution pipeline. Figure 1 now depicts the full conditioning mechanism at the convolutional level, while Figure 2 details the architectures of the modulation networks 𝑓 and 𝑔.
>
> ---
>
> ### 7. Figure quality and smoothing in Figure 4
>
> We agree that the images in Figure 4 appear smoother than typical diagnostic CTs. This is expected given the low projection count used in the cone-beam CT experiments (64 projections), which was chosen to simulate challenging low-dose and phase-resolved acquisition regimes. We have added a sentence clarifying this experimental setting to avoid confusion and changed the figure description.
>
> ---
>
> We thank again the reviewer for the careful reading and detailed feedback. We believe that the expanded discussion of related work, the clarified distinctions between conditioning mechanisms, and the additional experiments address the raised concerns and help better position the proposed approach.
>
>
> ---
>
> ### References
>
> [1] Lemay, A., Gros, C., Vincent, O., Liu, Y., Cohen, J. P., & Cohen-Adad, J. (2021). Benefits of linear conditioning with metadata for image segmentation. arXiv preprint arXiv:2102.09582.
>
> [2] Ramanarayanan, S., Murugesan, B., Palla, A., Ram, K., Venkatesan, R., & Sivaprakasam, M. (2023). MCI-HyperNet: A multiple contextual information-based adaptive weight learning network for controllable image reconstruction. Neurocomputing, 554, 126606.
>
> [3] Liu, X., Sanchez, P., Thermos, S., O’Neil, A. Q., & Tsaftaris, S. A. (2022). Learning disentangled representations in the imaging domain. Medical Image Analysis, 80, 102516.
>
> [4] https://docs.pytorch.org/docs/stable/generated/torch.nn.Conv2d.html

---

> ### Author Response · Authors · 2026-01-29
>
> We thank the reviewer again for the careful and constructive feedback. We hope that our responses and revisions address the raised points satisfactorily, and we remain available for any further clarification during the remainder of the discussion period.

---

### Author Rebuttal · Authors · 2026-01-25

**Rebuttal:**

We thank all reviewers for their careful assessment of our manuscript and for their constructive and detailed feedback. We appreciate the time and effort invested in reviewing our work and the thoughtful comments and suggestions provided. We have carefully considered all reviewer comments and revised the manuscript accordingly.

Where appropriate, we have added new experiments, clarified methodological choices, expanded the discussion of related work, and refined the presentation to better reflect the scope and contributions of the paper. Detailed, point-by-point responses addressing each concern are provided.

For the rebuttal, we have uploaded a single ZIP file containing the following materials:

- Revised Manuscript.pdf - the updated version of the paper
- Revised Manuscript with Highlighted Changes.pdf - the revised manuscript with all modifications clearly marked
- Responses to Reviewers.pdf - a detailed, point-by-point response addressing each reviewer’s comments individually

In addition, we have responded directly to each reviewer’s comments using the Official Comment feature in the review system.

We hope that the revisions and additional analyses address the reviewers’ concerns and strengthen the paper.

**Supporting Material:**

/attachment/67e486fdd0a0ab7b577b99b2a2c1a820a775ba11.zip

---

### Meta-Review · Area_Chair_iD94 · 2026-02-05

**Recommendation:** Accept (Poster)
**Confidence:** 4

**Metareview:**

Average score is 3.67. All reviewers found the proposed method to be novel and the results promising. Authors addressed the reviewers adequately.

---

### Decision · Program_Chairs · 2026-02-13

Accept (Poster)